# 3D mesh processing using GAMer 2 to enable reaction-diffusion simulations in realistic cellular geometries

**Christopher T. Lee**[1⊚], **Justin G. Laughlin**[1⊚], **Nils Angliviel de La Beaumelle**[1], **Rommie E. Amaro**[2], **J. Andrew McCammon**[2], **Ravi Ramamoorthi**[3], **Michael Holst**[4], **Padmini Rangamani**[1] *

**1** Department of Mechanical and Aerospace Engineering, University of California, San Diego, La Jolla, California, United States of America, **2** Department of Chemistry and Biochemistry, University of California, San Diego, La Jolla, California, United States of America, **3** Department of Computer Science and Engineering, University of California, San Diego, La Jolla, California, United States of America, **4** Department of Mathematics, University of California, San Diego, La Jolla, California, United States of America

⊚ These authors contributed equally to this work.
* prangamani@ucsd.edu

**Data Availability Statement:** The latest GAMer 2 code can be found on GitHub https://github.com/ctlee/gamer. Snapshots of GAMer 2 are also archived on Zenodo

## Abstract

Recent advances in electron microscopy have enabled the imaging of single cells in 3D at nanometer length scale resolutions. An uncharted frontier for *in silico* biology is the ability to simulate cellular processes using these observed geometries. Enabling such simulations requires watertight meshing of electron micrograph images into 3D volume meshes, which can then form the basis of computer simulations of such processes using numerical techniques such as the finite element method. In this paper, we describe the use of our recently rewritten mesh processing software, `GAMer 2`, to bridge the gap between poorly conditioned meshes generated from segmented micrographs and boundary marked tetrahedral meshes which are compatible with simulation. We demonstrate the application of a workflow using `GAMer 2` to a series of electron micrographs of neuronal dendrite morphology explored at three different length scales and show that the resulting meshes are suitable for finite element simulations. This work is an important step towards making physical simulations of biological processes in realistic geometries routine. Innovations in algorithms to reconstruct and simulate cellular length scale phenomena based on emerging structural data will enable realistic physical models and advance discovery at the interface of geometry and cellular processes. We posit that a new frontier at the intersection of computational technologies and single cell biology is now open.

## Author summary

3D imaging of cellular components and associated reconstruction methods have made great strides in the past decade, opening windows into the complex intracellular organization. These advances also mean that computational tools need to be developed to work with these images not just for purposes of visualization but also for biophysical

https://doi.org/10.5281/zenodo.2340294. High resolution versions of supplemental movies and meshes can be found at https://github.com/RangamaniLabUCSD/Lee-Laughlin-GAMer2. The EM data used in this work are from Wu, Y.; Whiteus, C.; Xu, C. S.; Hayworth, K. J.; Weinberg, R. J.; Hess, H. F.; Camilli, P. D. Contacts between the Endoplasmic Reticulum and Other Membranes in Neurons. PNAS 2017, 114 (24), E4859–E4867. https://doi.org/10.1073/pnas.1701078114. The original EM datasets are available by request from the corresponding author of this work, Pietro de Camilli (Pietro.decamilli@yale.edu), as pursuant to PNAS data availability guidelines.

**Funding:** CTL, REA, JAM, and MH are supported in part by the National Institutes of Health (https://www.nih.gov) under grant number P41-GM103426. CTL, and JAM are also supported by the National Institutes of Health (https://www.nih.gov) under RO1-GM31749. CTL also acknowledges support from the National institutes of Health (https://www.nih.gov) Molecular Biophysics Training Grant T32-GM008326 and a Hartwell Foundation Postdoctoral Fellowship. RR was supported in part by the Ronald L. Graham endowed chair. MH was supported in part by the National Science Foundation (https://www.nsf.gov) under awards DMS-CM1620366 and DMS-FRG1262982. PR was supported by the Air Force Office of Scientific Research (AFOSR, https://www.wpafb.af.mil/afrl/afosr/) Multidisciplinary University Research Initiative (MURI) FA9550-18-1-0051 and JGL was supported by a fellowship from the UCSD Center for Transscale Structural Biology and Biophysics/Virtual Molecular Cell Consortium (https://vmcc.ucsd.edu/). The funders had no role in study design, data collection and analysis, decision to publish, or preparation of the manuscript.

**Competing interests:** I have read the journals policy and the authors of this manuscript have the following competing interests: R.E.A. has equity interest in, is a cofounder of, and on the scientific advisory board of Actavalon, Inc.

simulations. In this work, we present our recently rewritten mesh processing software, `GAMer 2`, which features both mesh conditioning algorithms and tools to support simulation setup including boundary marking. Using a workflow that consists of other open-source software along with `GAMer 2`, we demonstrate the process of going from electron micrographs to simulations for several scenes of increasing length scales. In our preliminary finite element simulations of reaction-diffusion in the generated geometries, we reaffirm that the complex morphology of the cell can impact processes such as signaling. Technologies such as these presented here are set to enable a new frontier in biophysical simulations in realistic geometries.

This is a *PLOS Computational Biology* Methods paper.

## Introduction

Understanding structure-function relationships at cellular length scales (nm to $\mu$m) is one of the central goals of modern cell biology. While structural determination techniques are routine for very small and large scales such as molecular and tissue, high-resolution images of mesoscale subcellular scenes were historically elusive [1]. This was primarily due to the diffraction limits of visible light and the limitations of X-ray and Electron Microscopy (EM) hardware. Over the past decade, technological improvements such as improved direct electron detectors have enabled the practical applications of techniques such as volume electron microscopy [2–6]. Advances in microscopy techniques in recent years have opened windows into cells, giving us insight into cellular organization with unprecedented detail [7–13]. Volumetric EM enables the capture of 3D ultrastructural datasets (i.e., images where fine structures such as membranes of cells and their internal organelles are resolved, as shown in Fig 1A). Using these geometries as the basis of simulations provides an opportunity for the *in silico* animation of various cellular processes and the generation of experimentally testable hypotheses. Popular biophysical simulation modalities such as the finite element method [14, 15], particle-based stochastic dynamics [16–19], and the reaction-diffusion master equation [20–26] among many others require discretizations or meshes representing the geometry of the domain of interest. Moreover, in biological systems, the localization of molecular species is often heterogeneous [27, 28] which necessitates the need for boundary and region marking to represent this heterogeneity. To realize these simulations, therefore, a workflow is necessary to go from images to high-quality and annotated 3D meshes compatible with different numerical simulation modalities. As we review below, significant community effort has been invested in the imaging and segmentation steps, as well as mesh generation for graphics and visualization (Fig 1A–1C). In an effort to bridge advances in these fields, in this work we describe `GAMer 2`; software which takes input meshes generated from contour-tiling and segmentation, applies mesh conditioning algorithms from the graphics community, and marks faces to demarcate boundary conditions. `GAMer 2` is developed for the common biophysicist and features an easy to use user interface implemented as an add-on to 3D modeling software `Blender` along with a Python API `PyGAMer`. These user interfaces allow for the definition of marked boundaries corresponding to molecular localizations. The output of `GAMer 2` is a boundary marked and simulation compatible surface or volume mesh (Fig 1D and 1E) on which one can run finite element-based biophysical simulations (Fig 1F).

**Fig 1. Pipeline from electron microscopy data to a reaction-diffusion finite element simulation on a well-conditioned unstructured tetrahedral mesh.** A) Contours of segmented data overlaid on raw slices of electron microscopy data, B) Stacked contours from all slices of segmented data, C) Primitive initial 3D mesh reconstructed by existing IMOD software, D) Surface mesh after processing with our system; note the significantly higher quality of the mesh. The steps from C to D are the core contributions of this manuscript. Although this pipeline illustrates an application for images from electron microscopy, GAMer 2 is a general mesh conditioning library and can be used with meshes regardless of experimental context. E) Unstructured tetrahedral mesh suitable for finite element simulation obtained with TetGen software linked in GAMer 2, F) Reaction-diffusion model simulated using FEniCS software.

## Workflow steps from image to model

In order to develop models from image datasets, a series of steps must be executed. Starting with image acquisition and ending with a simulation compatible mesh, we summarize the typical workflow steps and highlight potential difficulties along the way.

**Image acquisition and segmentation.** Sample preparation begins with either cell culture or the harvesting of biological tissues of interest. Subsequent preparation steps can vary depending upon the particular volume EM imaging modality used but primarily include sample dehydration, fixation/staining, embedding, and imaging through the different cross-sections [29–32].

Once the images are captured, they are post-processed to improve properties such as contrast and alignment/registration across the stack. From the processed image stack, the boundaries of interesting features are traced or segmented. To the best of our knowledge, the state-of-the-art for segmenting electron micrographs of cells remains reliant upon the expertise of biologists for recognition of organelles and membrane domains in cells. During the segmentation process, the algorithm or researcher must carefully separate boundary signal from noise. Various schemes ranging from manual tracing, thresholding and edge-detection, to deep-learning based approaches have been employed to perform image segmentation [32, 33].

The resulting segmentations from volume EM can be visualized as stacks of contours (Fig 1B). This provides an initial glimpse into the 3D shapes of objects of interest. In order to enable modeling using the shapes represented by the contours, geometric meshes compatible with numerical methods can be constructed. However, a myriad of complexities often confound this process and necessitate flexible approaches for mesh generation.

**Meshing challenges.** A variety of challenges for meshing and subsequent physical simulations can arise at each step. Even with near-perfect experimental execution, and despite the enhanced surface contrast from heavy metal stains, images of cellular and organelle membranes are often poorly behaved and contain sharp and otherwise irregular geometries that are difficult to segment. In more serious cases, thinly sliced samples can tear or become contaminated during handling. Methodological errors are also possible. For example, Serial Block-Face Scanning Electron Microscopy (SBF-SEM) datasets in optimum conditions may have 3 nm lateral (x,y) resolution but 25 nm axial (z) resolution, limited by the slicing capability of the ultramicrotome [29]. Anisotropic resolution in tandem with variable slice thickness can cause loss of axial detail.

There are many EM software that post-process image stacks to correct for these and other artifacts. Most of our datasets have been manually segmented and corrected in software such as `IMOD` [34], `ilastik` [35], or `TrackEM2` [36]. `IMOD` and other tools such as `Contour-Tiler` in `VolRoverN` [37] among others [38, 39] have the capacity to perform contour-tiling operations to generate a preliminary surface mesh suitable for basic 3D visualization. If the end goal is visualizing the geometry of the cellular structure, then such meshing operations are often sufficient.

Meshes generated in this manner, however, are often not directly suitable for physical simulations due to various mesh artifacts as described below (Fig 1C). Some of these include jagged boundaries, non-manifold features, and high aspect ratio faces, as shown in Fig 2. These problems must be corrected to produce a conditioned surface mesh that is compatible with physical simulations (Fig 1D). For simulations that track concentrations in the volume, the conditioned surface mesh is tetrahedralized (Fig 1E). We note that although there exist advanced tetrahedral mesh generation tools, such as `TetWild` [40], and others [41–46], which can generate Finite Element Analysis (FEA) compatible volume meshes from these poor quality initial surfaces, these tools are general purpose and currently not adapted to the length scales of single cells and subcellular structures. Mesh defects such as disconnects in the Endoplasmic Reticulum (ER) arising from the limited resolving powers of EM or errors in segmentation (e.g., Fig 2C and 2D) require more careful and often manual curation. Currently, manual curation of cellular data sets remain the gold standard for identifying and matching cellular structures. A specialist trained in imaging modalities is capable of subjectively matching the identity of a surface with the underlying micrograph along with some history of observations from training to determine if an error is likely.

**Curating simulation metadata.** Once a suitable mesh is generated, other steps may be necessary to facilitate successful simulation. For example, when modeling biochemical signal transduction, receptors and other molecules may be localized to particular regions of a scene [47]. Realistic simulations must be able to represent the observed localization to effectively

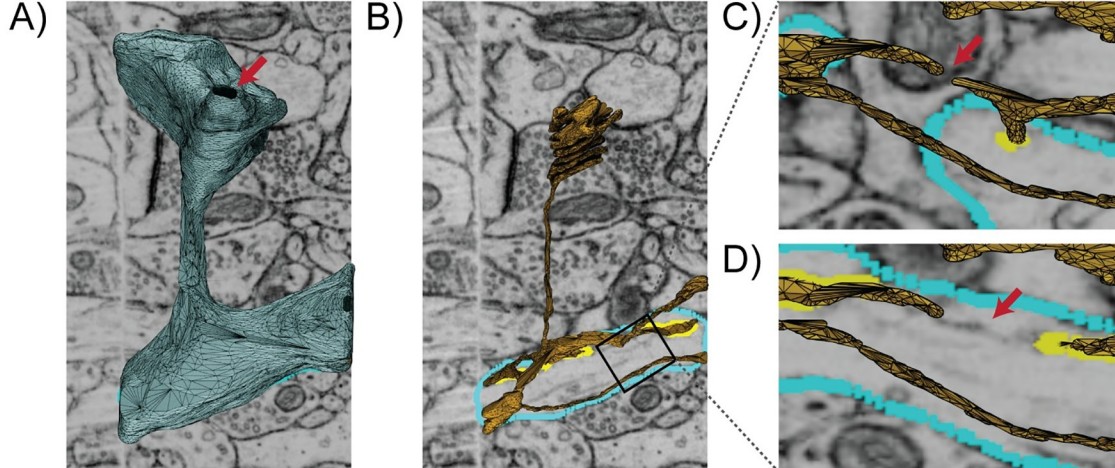

**Fig 2. Initial surface mesh model of a single spine scene with subcellular organelles imaged by Focused-ion Beam Milling Scanning Electron Microscopy (FIB-SEM) (overlaid), courtesy of Wu et al. [7], contains many mesh artifacts and is not compatible with physics-based simulations.** A) The blue surface represents the Plasma Membrane (PM) which contains a hole indicated by the red arrow. B) The yellow surface represents the membrane of the Endoplasmic Reticulum (ER). C, D) are two views rotated and zoomed-in on B showing a disconnected region of the ER proofed against micrographs taken at different z-axis locations. An untraced ghost, indicated by the red arrow, appears between the disconnected segments which suggests a possible error in the segmentation.

predict cellular behavior [48–50]. In a simulated model, the confinement of molecules can be presented as boundary conditions on a marked mesh region (Fig 1D). Depending on the situation, such localizations can be arbitrarily or randomly assigned for hypothesis testing [51] Alternatively, the regions of confinement may be informed by the electron micrographs themselves or correlated from another experimental approaches. A robust mesh generation tool capable of handing and resolving problems across all workflow steps including boundary marking and other metadata curation is necessary to support simulations from images of subcellular scenes (Fig 1E).

Here, we introduce our recently redesigned software, GAMer 2 (Geometry-preserving Adaptive MeshER version 2), which features mesh conditioning algorithms and simulation setup tools. In this redesign, we focused on the following software design criteria:

- Easy cross-platform code compilation and distribution.

- Runtime stability with meaningful error messages.

- Easy interactivity for the biophysicist user base.

- Version tracking for code provenance.

The algorithms in GAMer 2 for mesh conditioning remain those described by Yu *et al.* [52, 53], by Gao *et al.* [54, 55], and Chen and Holst [56]. As described in the original manuscripts, the algorithms seek to preserve mesh features while producing smooth surfaces. We will show in our illustrative examples that GAMer 2 approximately preserves volume. In this redevelopment, we have introduced the capability to perform local refinements and now provide end-users with access to new meta-parameters such as the number of neighbor rings to use in the calculation of the Local Structure Tensor (LST).

Details of the rewrite including the development of a new Python interface PyGAMer, and GAMer Blender add-on called BlendGAMer follow. In addition, we summarize new geometric capabilities such as the estimation of curvatures on meshes.

## Methods

### GAMer 2 development

GAMer 2 is a complete rewrite of GAMer in C++ using the CASC data structure [57] as the underlying mesh representation. Prior versions of GAMer were susceptible to segmentation faults under certain conditions, which is now fixed in this major update. In addition to improving the run-time stability, we have added error handling code to produce actionable notes for the convenience of the end-user. GAMer 2 continues to be licensed under LGPL v2.1 and the source code can be downloaded from GitHub (https://github.com/ctlee/gamer) [58].

In this update, we have also redeveloped the Python interface for GAMer and the Blender add-on. The GAMer 2 Python API, called PyGAMer is generated using pybind11 [59] as opposed to SWIG [60] used by GAMer. pybind11 provides superior wrapping of C++ template objects which enables PyGAMer to interact with nearly all elements of GAMer 2 in a Python environment. The GAMer 2-Blender add-on, now called BlendGAMer, has been rewritten to use the PyGAMer interface. Moreover, the latest BlendGAMer release, v2.0.6, supports Blender versions 2.79b, and 2.8X. Blender not only provides a customizable mesh visualization environment, but also tools such as sculpt mode, which allows users to flexibly manipulate the geometry [61]. With great care to remain truthful to the underlying data, Blender sculpting tools can be useful for fixing topological issues such as disconnected ER

segments. We briefly review the concepts behind the mesh processing algorithms from Yu et al. [52, 53].

## Mesh processing

**Local structure tensor.**  The mesh processing operations in `GAMer` are designed to improve mesh quality while preserving the underlying geometry of the data. We use a LST to account for the local geometry [62–64]. The LST is defined as follows,

$$T(\mathbf{v}) = \sum_{i=1}^{N_r} \mathbf{n_i} \otimes \mathbf{n_i} = \sum_{i=1}^{N_r} \begin{pmatrix} n_i^x n_i^x & n_i^x n_i^y & n_i^x n_i^z \\ n_i^y n_i^x & n_i^y n_i^y & n_i^y n_i^z \\ n_i^z n_i^x & n_i^z n_i^y & n_i^z n_i^z \end{pmatrix}, \tag{1}$$

where $\mathbf{v}$ is the vertex of interest, $N_r$ is the number of neighbors in the $r$-ring neighborhood, and $n_i^{x,y,z}$ form the normal of the $i$th neighbor vertex. Vertex normals are defined as the weighted average of incident face normals. Performing the eigendecomposition of the LST, we obtain information on the principal orientations of normals in the local neighborhood [65]. The magnitude of the eigenvalue corresponds to the amount of curvature along the direction of the corresponding eigenvector. Inspecting the magnitude of the eigenvalues gives several geometric cases:

- Planes: $\lambda_1 \gg \lambda_2 \approx \lambda_3 \approx 0$

- Ridges and valleys: $\lambda_1 \approx \lambda_2 \gg \lambda_3 \approx 0$

- Spheres and saddles: $\lambda_1 \approx \lambda_2 \approx \lambda_3 > 0$

**Feature preserving mesh smoothing.**  Finite element simulations are sensitive to the mesh quality [66, 67]. Poor quality meshes can lead to numerical error, instability, long times to solution, and non-convergence. Generally, triangular meshes with high aspect ratios produce larger errors compared with equilateral elements [68].

To improve the conditioning of the surface meshes derived from microscopy images, we use an angle-weighted Laplacian smoothing approach, as shown in Fig 3A. This scheme is an extension of the angle weighted smoothing scheme, formulated for 2D and described by Zhou and Shimada, to three dimensions [69]. In essence, this algorithm applies local torsion springs to the 1-ring neighborhood of a vertex of interest to balance the angles.

Given a vertex $\mathbf{x}$ with the set of 1-ring neighbors $\{\mathbf{v}_1, \ldots, \mathbf{v}_N\}$, where $N$ is the number of neighbors, ordered such that $\mathbf{v}_i$ is connected to $\mathbf{v}_{i-1}$ and $\mathbf{v}_{i+1}$ by edges. The 1-ring is connected such that $\mathbf{v}_{N+1} := \mathbf{v}_1$ and $\mathbf{v}_{-1} := \mathbf{v}_N$. Traversing the 1-ring neighbors, we define edge vectors $\mathbf{e}_{i-1} := \overrightarrow{\mathbf{v}_i \mathbf{v}_{i-1}}$ and $\mathbf{e}_{i+1} := \overrightarrow{\mathbf{v}_i \mathbf{v}_{i+1}}$. This algorithm seeks to move $\mathbf{x}$ to lie on the perpendicularly bisecting plane $\Pi_i$ of $\angle(\mathbf{v}_{i-1}, \mathbf{v}_i, v_{i+1})$. For each vertex in the 1-ring neighbors, we compute the perpendicular projection, $\mathbf{x}_i$, of $\mathbf{x}$ onto $\Pi_i$. Since small surface mesh angles are more sensitive to change in $\mathbf{x}$ position than large angles, we prioritize their maximization. We define a weighting factor, $\alpha_i = \frac{\mathbf{e}_{i-1} \cdot \mathbf{e}_{i+1}}{|\mathbf{e}_{i-1}| \cdot |\mathbf{e}_{i+1}|}$, which inversely corresponds with $\angle(\mathbf{v}_{i-1}, \mathbf{v}_i, \mathbf{v}_{i+1})$. The average of the projections weighted by $\alpha_i$ gives a new position of $\mathbf{x}$ as follows,

$$\bar{\mathbf{x}} = \frac{1}{\sum_{i=1}^{N}(\alpha_i + 1)} \sum_{i=1}^{N} (\alpha_i + 1)\mathbf{x}_i. \tag{2}$$

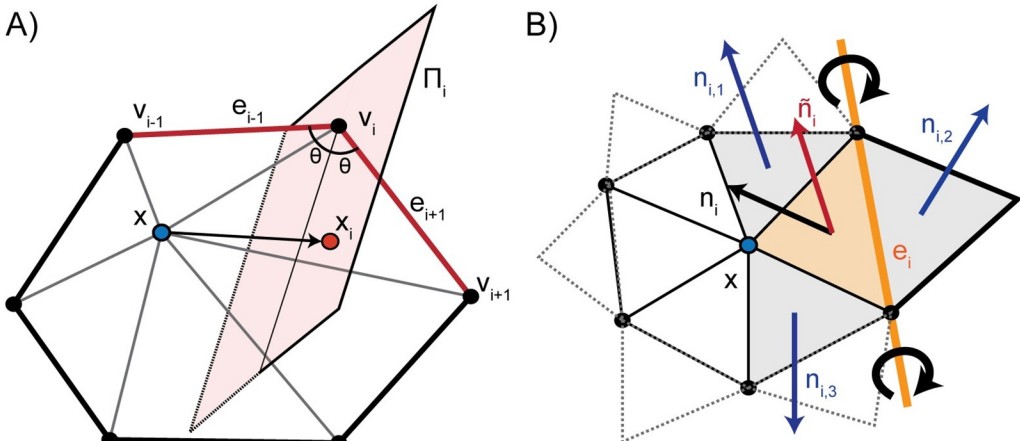

**Fig 3. Schematic illustrating `GAMer` mesh conditioning algorithms.** A) angle-based surface mesh conditioning, and B) anisotropic normal smoothing algorithms which are previously described by Yu et al. [52, 53] and implemented in `GAMer`.

There are many smoothing algorithms in the literature; the angle-weighted Laplacian smoothing algorithm described here can outperform other popular smoothing strategies such as those described in [70–73] which are primarily focused on optimizing the smoothness of surface normals for computer graphics applications and not mesh angles. Our goal is not to provide an elaborate comparison against existing algorithms in this manuscript but to demonstrate the utility of our pipeline for biological images, with a specific goal of using EM-generated images for computational biology simulations.

Conceptually the fidelity of the local geometry can be maintained by restricting vertex movement along directions of low curvature. This constraint is achieved by anisotropically dampening vertex diffusion using information contained in the LST. Although the weighted vertex smoothing scheme, as described, will reasonably preserve geometric structure, the structure preservation can be further improved by using the LST. Computing the eigendecomposition of the LST, we obtain eigenvalues $\lambda_1, \lambda_2, \lambda_3$ and eigenvectors $\mathbf{E}_1, \mathbf{E}_2, \mathbf{E}_3$, which correspond to principal orientations of local normals. We project $\bar{\mathbf{x}} - \mathbf{x}$ onto the eigenvector basis and scale each component by the inverse of the corresponding eigenvalue,

$$\hat{\mathbf{x}} = \mathbf{x} + \sum_{k=1}^{3} \frac{1}{1 + \lambda_k} [(\bar{\mathbf{x}} - \mathbf{x}) \cdot \mathbf{E}_k] \mathbf{E}_k. \tag{3}$$

This has the effect of dampening movement along directions of high curvature i.e., where $\lambda$ is large. In this fashion, our algorithm improves triangle aspect ratios while preserving local geometric features. We note that our actual implementation iterates between rounds of vertex smoothing and conventional angle based edge flipping to achieve the desired smoothing effect. Edge flips are common in mesh processing, and provide a mechanism for both improving angles and reducing the valency of vertices [74]. A comparison of the angle-weighted smoothing algorithm with and without LST correction is shown in Fig 4.

**Feature preserving anisotropic normal-based smoothing.** To remove additional bumpiness from the mesh, we use a normal-based smoothing approach [75, 76], as shown in Fig 3B. The goal is to produce smoothly varying normals across the mesh without compromising mesh angle quality. Given a vertex $\mathbf{x}$ of interest, for each incident face $i$, with normal $\mathbf{n}_i$ we rotate $\mathbf{x}$ around a rotation axis defined by opposing edge $e_i$ such that $\mathbf{n}_i$ aligns with the mean

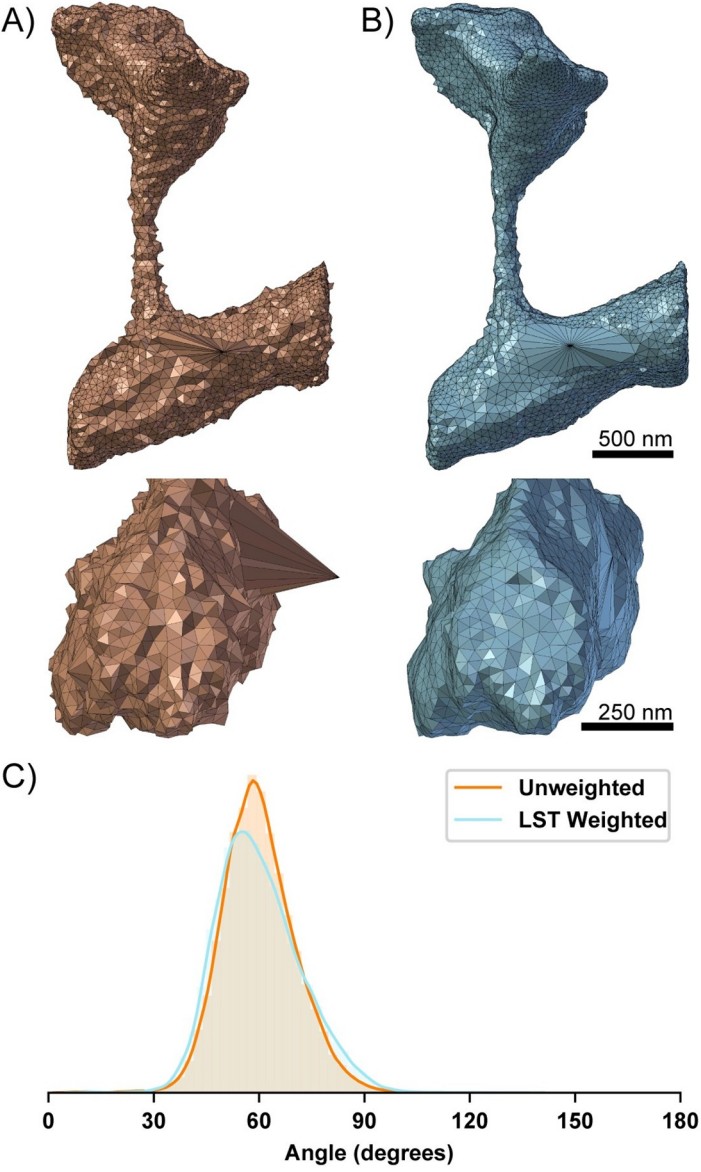

**Fig 4. Comparison of 50 iterations of angle weighted smoothing algorithm.** A) without and B) with Local Structure Tensor (LST) based correction. The LST helps to preserve the geometric structure albeit with slight degradation to the mesh angles. It is a simple metric to capture local geometric information which can be used to constrain conditioning operations. C) Mesh angles are improved in both the LST weighted and unweighted meshes. The distribution of the mesh angles with LST correction are left shifted from 60 degrees.

normal of neighboring faces $\bar{\mathbf{n}}_i = \sum_{j=1}^{3} \mathbf{n}_{ij}/3$. We denote the new position which aligns $\mathbf{n}_i$ and $\bar{\mathbf{n}}_i$ as $R(x; e_i, \theta_i)$. Summing up the rotations and weighting by incident face area, $a_i$, we get an updated position,

$$\bar{\mathbf{x}} = \frac{1}{\sum_{i=1}^{N_1} a_i} \sum_{i=1}^{N_1} a_i R(\mathbf{x}; e_i, \theta_i). \tag{4}$$

This is an isotropic scheme that is independent of the local geometric features; meaning that many iterations of this algorithm may weaken sharp features.

Instead, we use an anisotropic scheme [76, 77] to compute the mean neighbor normals,

$$\bar{\mathbf{n}}_i = \frac{1}{\sum_{j=1}^{3} e^{K(\mathbf{n}_i \cdot \mathbf{n}_{ij})}} \sum_{j=1}^{3} e^{K(\mathbf{n}_i \cdot \mathbf{n}_{ij})} \mathbf{n}_{ij}, \tag{5}$$

where $K$ is a user defined positive parameter which scales the extent of anisotropy. Under this scheme, the weighting function decreases as a function of the angle between $\mathbf{n}_i$ and $\mathbf{n}_{ij}$ resulting in the preservation of sharp features.

**Feature preserving mesh decimation.**   The number of degrees of freedom in the mesh influences the computational burden of subsequent physical simulations. One strategy to reduce the number of degrees of freedom is to perform mesh decimation or simplification.

There are many strategies for decimation, some reviewed here [78, 79], including topology preserving Euler operators, other algorithms such as vertex clustering which may not guarantee topological invariance [80], and remeshing [81]. It is typically desirable to preserve the mesh topology for physical simulation based applications. Conventional Euler operations for mesh decimation include vertex removal, edge collapse, and half-edge collapse. As noted earlier, finite element simulations are sensitive to angles of the mesh. Edge and half-edge collapses can sometimes lead to vertices with high or low valency and therefore poor angles. Although algorithms to detect topology-changing edge collapses have been developed [82], we avoid this problem by employing a vertex removal algorithm. First, vertices to be decimated are selected based upon certain criteria discussed below. We then remove the vertex and re-triangulate the resulting hole. This is achieved using a recursive triangulation approach, which heuristically balances the edge valency. Given the boundary loop, we first connect vertices with the fewest incident edges. This produces two resulting holes that we then fill recursively using the same approach. When a hole contains only three boundary vertices, they are connected to make a face. We note that while this triangulation scheme balances vertex valency, it may degrade mesh quality. We solve this by running the geometry preserving smoothing algorithm on the local region.

We employ two criteria for selecting which vertices to remove. These criteria can be used in isolation or together. First, to selectively decimate vertices in low or high curvature regions, information from the LST can be used. Comparing the magnitudes of the eigenvalues of the LST provides information about the local geometry near a vertex. For example, to decimate vertices in flat regions of the mesh, given eigenvalues $\lambda_1 \geq \lambda_2 \geq \lambda_3$, vertices can be selected by checking if the local region satisfies,

$$\frac{\lambda_2}{\lambda_1} < R_1, \tag{6}$$

where $R_1$ is a user specified flatness threshold (smaller is flatter). In a similar fashion, vertices in curved regions can also be selected. However, decimation of curved regions is typically avoided due to the potential for losing geometric information.

Instead, to simplify dense areas of the mesh, we employ an edge length based selection criterion,

$$\frac{\max_{i=1}^{N_1} d(\mathbf{x}, \mathbf{v}_i)}{\bar{D}} < R_2, \tag{7}$$

where $N_1$ is the number of vertices in the 1-ring neighborhood of vertex $\mathbf{x}$, $d(\cdot, \cdot)$ is the distance between vertices $\mathbf{x}$ and $\mathbf{v}_i$, $\bar{D}$ is the mean edge length of the mesh, and $R_2$ is a user

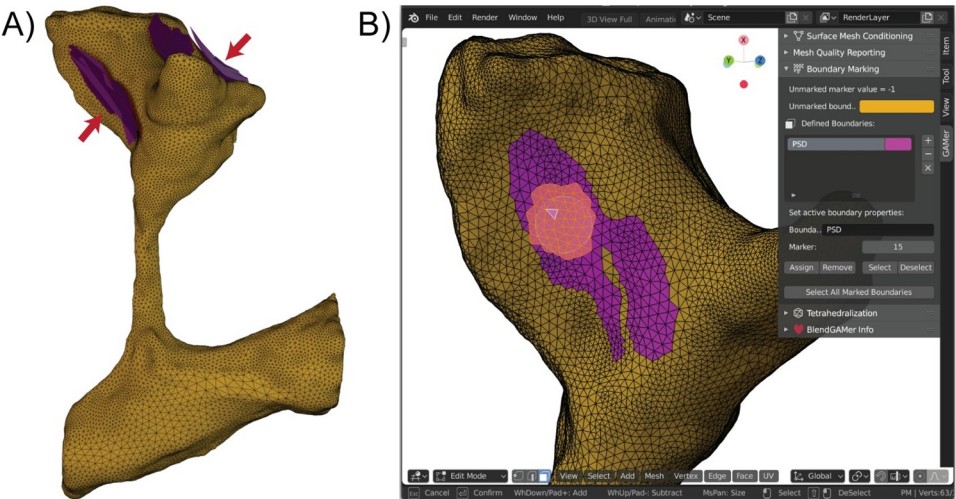

**Fig 5. Marking boundaries using `BlendGAMer`.** A) The Postsynaptic Density (PSD) is annotated as a contour in the segmentation and represented as a patch (purple) neighboring the plasma membrane (yellow). B) Screenshot of boundary marking using `BlendGAMer` v2.0.5 in `Blender` 2.80. The circle select tool is used to select faces of the plasma membrane mesh in proximity to the PSD patches. A user interface allows the user to name boundaries, assign and unassign face membership, along with setting the marker value. At the time of writing, `BlendGAMer` runs in `Blender` versions ranging from 2.79b and onward.

specified threshold. This criterion allows us to control the sparseness of the mesh. We note that the aforementioned criteria are what is currently implemented in `GAMer 2`, however the vertex removal decimation scheme can be employed with any other selection criteria.

**Boundary marking and tetrahedralization.** To support the definition of boundary conditions on the mesh, it is conventional to assign boundary marks or identifiers which correspond to different boundary definitions in the physical simulation. In simplified and idealized geometries it is possible to define functions to assign boundary values. However, in subcellular scenes where the geometry may be tortuous, local receptor clusters can be arbitrarily distributed on the manifold, and the resolution may be insufficient to resolve these features, boundary definition is a non-trivial challenge. In many scenarios, the most biologically accurate boundary definition may be based off of a biologist's understanding of the specimen which transcends the particular datset. For example, a particular image may not be able to resolve how receptors are distributed but knowledge of relevant immunogold labeling studies may allow the biologist to propose a physiologically meaningful receptor distribution. The `Blend-GAMer` add-on supports the facile user-based definition of boundary markers on the surface [61]. Users can utilize any of the face selection methods (e.g., circle selection demonstrated in Fig 5) which `Blender` provides to select boundaries to mark. Boolean operations and other geometric strategies provided natively in `Blender` can also be used for selection. Boundary markers are associated with a unique material property which helps visually delineate marked assignments. After boundaries are marked, stacks of surface meshes corresponding to different domains can be grouped and passed from `GAMer 2` into `TetGen`, which generates a volume mesh by constrained Delaunay tetrahedralization [83]. Each surface mesh can also be assigned a region marker, which is used by `TetGen` to assign marker values for the enclosed tetrahedra.

## Results

As a demonstration, we apply `GAMer 2` to build simulations from electron micrographs and segmentations from Wu et al. [7] which were graciously shared by De Camilli and coworkers. In their work, Wu et al. imaged dendritic spines from neurons taken from the mouse cerebral cortex or nucleus accumbens using Focused-ion Beam Milling Scanning Electron Microscopy (FIB-SEM) [7]. In addition to their important role in synaptic and structural plasticity, these cellular structures demonstrate highly tortuous morphologies, high surface-to-volume ratios, and a geometric intricacy that serves as a good test-bed for our approach. Here we consider several scenes of increasing length scale: the ER of the single spine geometry which requires nanometer precision and the Plasma Membrane (PM) of the single spine which has a length scale of a couple of microns (Fig 6A), the two spine geometry, a few microns (Fig 6B), and the dendrite with about 40 spines, with a length scale in the tens of microns (Fig 6C). For each of these geometries, using the segmentations produced by Wu et al., we generated preliminary meshes using the `imod2obj` utility included with `IMOD` [34]. The output initial meshes have units of pixels and were scaled to nm using an 8 nm isotropic voxel size. Each initial mesh was then processed using algorithms described in §*Mesh processing* and implemented in `GAMer 2` [58]. We note that for some meshes, features such as disconnections of the ER, were manually reconnected using `Blender` mesh sculpting features. We will provide a candid discussion of the manual curation steps in the following paragraphs discussing each scene. Boundaries were marked using `BlendGAMer`, although `PyGAMer` and `GAMer 2` can be scripted to assign boundary labels as well, and the conditioned surface meshes were tetrahedralized using `TetGen` [83].

The one spine geometry contains two separate membranes: the PM and the ER Fig 6A—each with their own problems. The PM contained several large holes on the surface which correspond to the top and bottom of the image stack. As the dendrite meanders throughout the tissue block, the cutting planes of the experiment or sample block may result in the artificial truncation of the image. We have remedied this truncation by triangulating the holes and processing using `GAMer 2` algorithms.

The ER mesh contained many more initial artifacts. This is because the detailed and variable nature of the ER membrane can be poorly resolved by the imaging method. Nixon-Abel and coworkers found using superresolution fluorescence microscopy that ER tubules have a diameter of 50 to 100 nm and sheet-like structures at the cellular periphery can be much finer [84]. Some ER morphology cannot even be resolved by the powers of EM [7]. For example, if the ER undergoes large spatial variation between z-slices then tubules may appear disconnected. Alternatively, the boundaries of the ER membrane may have poor contrast and can sometimes be missed during segmentation. We have manually reconnected the ER segments in `Blender` under the guidance of the underlying EM micrographs. An animated comparison of the initial mesh with the stack of images is shown in S1 Movie. This type of topological artifact arising from errors in segmentation or poor imaging resolution remains a major challenge to the field. In this work, to produce a model faithful to the biological context, we have employed human judgement to detect and rectify similar problems. As imaging technology and machine recognition algorithms improve, we anticipate that the need for manual curation will be reduced. The decision to manually curate, or not, is at the discretion of the end-user. `BlendGAMer` provides only the option for the tight integration of sculpting and automated mesh processing.

After curation and processing with `GAMer 2`, the geometric detail of the one spine scene is preserved. An animated comparison of the meshes output by `GAMer 2` is shown in S2 Movie. Notably, this spine contains a specialized form of ER termed the spine apparatus, Fig 6A, inset,

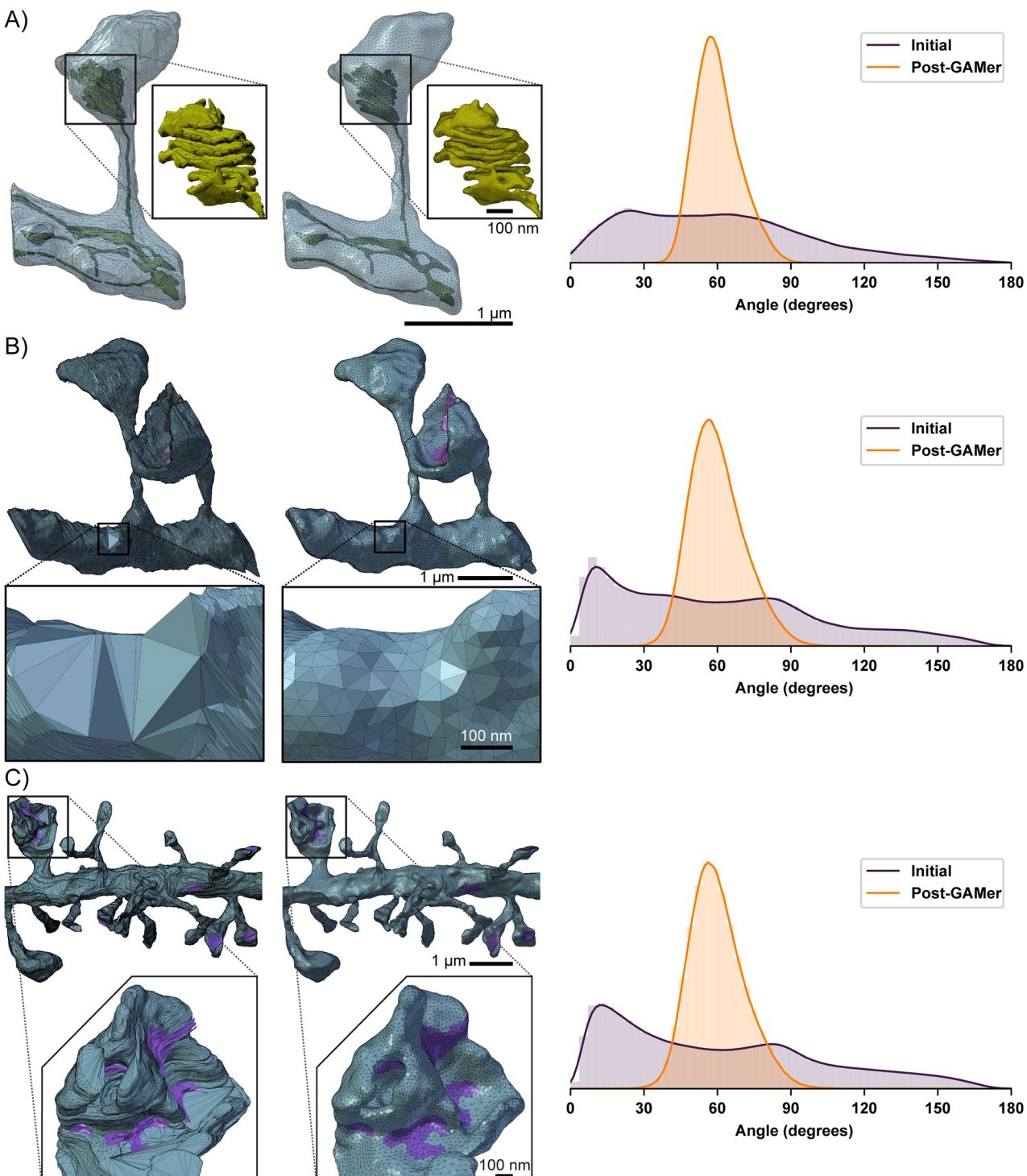

**Fig 6. Quantification of mesh quality pre- and post-`GAMer 2` processing for several geometries.** Data at varying spatial scales can be processed via the `GAMer 2` framework. Surface meshes of dendritic spine geometries before (left) are compared with their mesh after `GAMer 2` processing (middle). The shift in distribution of angles highlights the improvement in mesh quality (right). A) Surface meshes of a single dendritic spine; the PM is colored cyan and the ER yellow. Inlay: close-up of the spine apparatus. The standard deviation of the distribution of triangular angles before is 35.9 and after 9.1. B) Surface mesh of PM of two dendritic spines. Faces marked as purple are the PSD. Inlay: close-up of a region with a large variance in angle distribution before `GAMer 2` processing. The standard deviation of the distribution of triangular angles before is 41.1 and after 11.1. C) Surface mesh of PM of a dendrite segment with many spines. Inlay: `GAMer 2` preserves the intricate details of a highly curved spine head with multiple regions of PSD. The standard deviation of the distribution of triangular angles before is 41.9 and after 11.1.

which consists of seven folded cisternae. This highly organized structure bears geometrical similarities to a parking garage structure and helicoidal geometries [85–88]. The geometric detail of the spine apparatus is preserved by the conditioning process in our pipeline.

In Fig 6, we also show the distribution of the triangular angles of the surface mesh before and after conditioning. One metric of a well-conditioned mesh is that all the surface triangles are nearly equilateral [68]. Prior to conditioning, the angle distribution is spread out and contains many large and small angles. After processing using `GAMer 2`, the angles of triangles of the one spine PM mesh are improved, as indicated by the peaked distribution around 60 degrees. Although the ER structure is significantly more complex, the angles of triangles of the mesh are also improved, albeit to a lesser extent than the PM. In scenarios such as this where the length scales of interest are closer to the acquisition resolution, it may be necessary to increase the number of triangles to accurately capture the fine details with high mesh quality. Table 1 summarizes the number of vertices and triangles in the initial vs conditioned meshes as well as vertices and tetrahedra in the resultant volumetric meshes. To accurately capture the curvature of the PM mesh in Fig 6A about 48% more triangles were needed compared to the ER mesh in Fig 6A, inset, which required 270% more triangles, both relative to the initial mesh.

The approach described here is also applicable for larger systems as we demonstrate with two spines and a full dendrite. The two spine geometry shown in Fig 6B is a few microns in length. Based on the length scales we would expect a well conditioned mesh for this geometry to contain approximately double the number of triangles in the single spine mesh; however, the orientation of z-stacks in this mesh is different from that in the single spine geometry which led to an abnormally large number of triangles: 320,976 versus just 9,330 in the mesh of PM in the single spine. After `GAMer 2` conditioning algorithms were applied, the number of triangles was reduced to 36,050, a much more reasonable count. As demonstrated, our pipeline is robust and can handle cases where the initial mesh either generates too few or too many triangles as required for capturing geometric details. The ER of the two spine geometry was processed in a similar manner to that of the one spine.

At the tens of microns length scale, we constructed a mesh of a full dendritic segment. We show a zoomed in section of the mesh before and after conditioning in Fig 6C. As in the one and two spine cases, our system robustly handles artifacts such as poor quality triangles and intersecting faces; Fig 6C shows that the distribution of the angles post conditioning are comparable to the one and two spine examples, showing that size does not alter the capability to produce well-conditioned meshes. Fig 6C shows an intricate spine head with many different regions of PSD shown in purple; this geometry is preserved post-conditioning and the PSD is marked with `BlendGAMer` to denote a boundary condition.

For all meshes the initial surface area is greater than that of the final result (Table 1. This is due to the jagged nature of the initial meshes which reflects small deviations in the alignment and registration of the micrographs and segmentation. As the surfaces are smoothed, the surface area is therefore reduced. On the other hand, the initial and final volumes of each geometry remain similar. This is a good indication of the feature-preserving nature of the algorithms.

In Fig 7 we compare the meshes generated by `GAMer 2` with other software including `TetWild` [40], `CGAL 3D Mesh Generation` (referred to as `CGAL` in the subsequent text) [44–46], Hu et al. Remesh [89], and `VolRoverN` [37]. For this analysis, we performed a best faith effort to use these tools using recommended default settings where possible, additional details are provided in S1 Appendix. We do not make any claims of software supremacy and instead highlight feature differences between the codes. Shown in Fig 7A are the distribution of triangular angles for the final surface mesh. We find that `GAMer 2` produces a mesh with more equilateral triangles than other tools.

**Table 1. Vertex and element counts for meshed geometries.**

|  |  | Surface Mesh | | | Volume Mesh* | | |
| --- | --- | --- | --- | --- | --- | --- | --- |
|  |  | # Vertices | # Triangles | Area [$\mu m^2$] | # Vertices | # Tetrahedra | Volume [$5\mu m^3$] |
| **Single Spine** | Initial PM | 4,695 | 9,330 | 6.99 | 6,726 | 27,581 | 0.64 |
|  | Conditioned PM | 6,924 | 13,844 | 6.52 | 13,734 | 62,557 | 0.65 |
|  | Initial ER | 6,546 | 19,654 | 2.70 | – | – | 0.028* |
|  | Conditioned ER | 36,294 | 72,620 | 2.39 | 53,134 | 211,018 | 0.027 |
| **Two Spines** | Initial PM | 160,733 | 320,976 | 26.09 | – | – | 2.82* |
|  | Conditioned PM | 18,027 | 36,050 | 23.58 | 28,989 | 122,082 | 2.94 |
|  | Initial ER | 20,111 | 40,370 | 10.90 | – | – | 0.13* |
|  | Conditioned ER | 80,419 | 161,050 | 9.73 | 101,033 | 352,741 | 0.13 |
| **Dendritic Segment** | Initial PM | 207,448 | 410,896 | 139.1 | – | – | 10.403* |
|  | Conditioned PM | 126,336 | 252,668 | 110.0 | 194,848 | 798,626 | 10.611 |

*Non-manifold and other mesh artifacts prevent the tetrahedralization of these meshes. Volumes reported are computed using the corresponding surface mesh.

The radius-ratio is a useful metric for determining the quality of both triangles and tetrahedra, it is defined as $\frac{nr_i}{r_o}$ where $n$ is the geometric dimension (i.e. 2 for triangles, 3 for tetrahedra), $r_i$ is the radius of the largest $n$-sphere which can be inscribed within the shape, and $r_o$ is the radius of the smallest $n$-sphere which circumscribes the shape. An equilateral triangle and an equilateral tetrahedron both correspond to a radius-ratio of one [90]. As the radius-ratio approaches zero the element approaches degeneracy which can affect numerical accuracy, stability, and convergence [66, 67].

Fig 7B compares different codes and their resultant distributions of radius-ratios when applied to the single spine PM surface mesh. One important difference is that TetWild, CGAL, and Hu et al. Remesh are fully automated algorithms which employ constraints to guarantee preservation of input features. We note that all methods tested here, except TetWild, require a watertight and manifold surface mesh as input. For the sake of this comparison, the same manually curated watertight and manifold meshes were used as input for all methods. Noisy features such as the jagged boundaries resulting from misaligned micrographs or segmentation are often preserved in the surface meshes generated by these codes. As a result, the triangular angles often deviate from equilateral in order to represent these preserved fine features. GAMer 2 on the other hand does not strictly preserve the fine features of the input mesh. The LST is a metric of the local curvature averaged over a mesh patch. Thus, GAMer 2 generated meshes are smoother but deviate, from the input, more than those generated by TetWild, CGAL, and Hu et al. Remesh. VolRoverN, on the other hand, implements the Level Set Boundary-Interior-Exterior (LBIE) algorithm which employs a geometric flow. The LBIE algorithm is known to work well for spherical geometries and was designed for smoothing biomolecular meshes constructed as the union of hard spheres [37, 91, 92]. Given the complex geometry of the dendritic spine, VolRoverN does not preserve the geometry well.

The tetrahedral mesh qualities of meshes produced by each code are compared in Fig 7C and 7D. Notably, no volume meshes are shown for Hu et al. Remesh and VolRoverN since Hu et al. is a surface remeshing code only and VolRoverN failed to produce a valid volume mesh when called from the software's graphical user interface. The other codes TetWild, CGAL, and GAMer 2 all produced quality tetrahedral meshes as output. The distribution of tetrahedral radius-ratios are shown in Fig 7D. We find that CGAL under-performs in

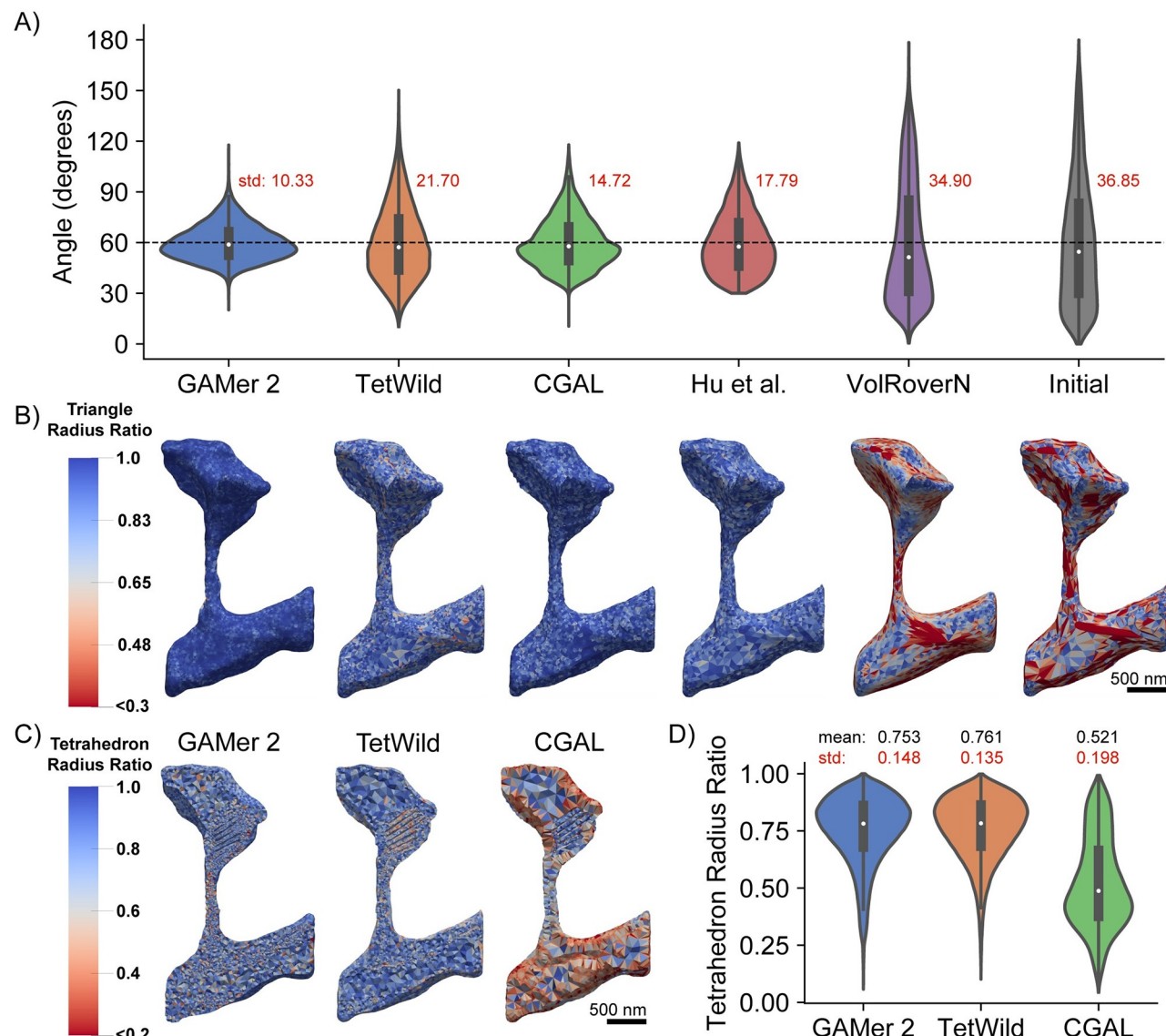

**Fig 7. Comparison of `GAMer 2` generated meshes with leading mesh generation software.** A) Distribution of triangular angles of the final Plasma Membrane (PM) and Endoplasmic Reticulum (ER) surface meshes. B) Quantification of the surface mesh triangle radius-ratios. The order of meshes is identical to the software ordering in panel A. C) Quantification of tetrahedron radius-ratios of the tetrahedral mesh output by each software. We note that the tetrahedral mesh produced by `GAMer 2` is generated using `TetGen` from a `GAMer 2` conditioned surface mesh. Also, in our best faith effort, we were not able to tetrahedralize the PM and ER mesh using `VolRoverN`. D) Distribution of tetrahedron radius-ratios of the resulting tetrahedral meshes.

comparison to the other two codes. This observation may be due, in part, to our usage of `CGAL`. The 3D mesh generation subproject of `CGAL` is provided as a library along with several example C++ scripts. For this comparison, we have applied the bundled script to our mesh of interest with minimal modification. We note that the default settings in the script may not be ideal for our application. It is possible that by setting stricter mesh quality targets for optimization, `CGAL` could perform better.

## GAMer 2 operations preserve mesh topology

As aforementioned, where we deemed appropriate, manual curation was used to modify the topology of the input meshes. In order to differentiate between the manual modifications and to verify whether GAMer 2 operations change the topology of the mesh we have computed Betti numbers at several mesh processing stages. Betti numbers are topological invariants and thus can be used to distinguish between topological spaces. The $k$th Betti number $\beta_k$ describes the number of $k$-dimensional holes on a surface. Intuitively $\beta_0$ is the number of connected components, $\beta_1$ is the number of circular holes, and $\beta_2$ is the number of enclosed voids.

The computation of Betti numbers is performed using a modified algorithm by Definado-Edelsbrunner [93]. In their original work, they describe an algorithm to compute the Betti numbers of tetrahedral meshes by iterating over a filtration. Since GAMer 2 is primarily a surface mesh conditioning library, there are no tetrahedra and therefore the Delfinado-Edelsbrunner algorithm is not applicable. The challenge lies in the determination of when the addition of a triangular face (i.e., 2-simplex) to a filtration forms an enclosed volume. In the original algorithm, the tetrahedral simplices are colored such that all adjacent tetrahedra not separated by a closed boundary will be the same color. The addition of a triangular face in a filtration will produce a void if and only if it completes a boundary such that the interior and exterior tetrahedra can be colored differently. While algorithms such as flood-filling or ray-casting and counting surface crossings have been described, these approaches are often computationally cumbersome especially when applied to meshes with reentrant surfaces.

We simplify the problem at hand by restricting our calculation to apply for the set of topologically manifold surface meshes with or without boundaries. If a mesh is both orientable (i.e., a consistent normal orientation can be assigned for all faces such that no neighboring faces have opposing normals) and all edges belong to two faces, we say that the mesh is a closed manifold. After iterating through a filtration in the manner described by Delfinado-Edelsbrunner, if we find that a mesh is a closed manifold then we increment the first and second Betti numbers $(\beta_1, \beta_2)$ by one. If the mesh is not a closed manifold, then no action is taken and the algorithm terminates and reports the first three Betti numbers. At any point iterating through the filtration, if an edge is found to participate in three or more faces, the mesh is not topologically manifold and only the zeroth Betti number is reported.

Betti numbers of the meshes at several processing stages, produced by the modified Delfinado-Edelsbrunner algorithm, are computed and shown in Table 2. As aforementioned, the initial meshes produced by imod2obj often contain disjoint surfaces or holes. Comparing against the underlying micrographs and data, we have curated each mesh to produce a watertight model. As an alternative to manual curation, other approaches such as persistent homology can be used to filter out defects [94]. At this point we apply GAMer 2 meshing operations to produce a conditioned mesh. We find that the topology of the watertight meshes are identical to that of their corresponding conditioned mesh.

## Estimating membrane curvatures in realistic geometries

In addition to the generation of simulation compatible meshes of realistic cell geometries, the meshes from GAMer 2 are amenable to other geometric analysis. The conditioned meshes can yield improved results for many geometric quantities of interest such as surface area and volume along with other more complex observables such as surface curvature. Membrane curvatures and minimal surfaces have long been of interest to biophysicists and mathematicians alike.

One of the advantages of using electron micrographs of membrane structures in cells is that we can now bridge the gap between membrane mechanics, curvature studies, and realistic

**Table 2. Computed Betti numbers for each mesh.**

| Geometry | Component | $\beta_0$ | $\beta_1$ | $\beta_2$ |
|---|---|---|---|---|
| Single Spine | Initial PM† | 1 | 4 | 0 |
| | Watertight PM* | 1 | 0 | 1 |
| | Conditioned PM | 1 | 0 | 1 |
| | Initial ER† | 6 | 18 | 6 |
| | Watertight ER* | 1 | 18 | 1 |
| | Conditioned ER | 1 | 18 | 1 |
| Two Spines | Initial PM† | 1 | 0 | 1 |
| | Watertight PM* | 1 | 0 | 1 |
| | Conditioned PM | 1 | 0 | 1 |
| | Initial ER† | 3 | 110 | 3 |
| | Watertight ER* | 1 | 110 | 1 |
| | Conditioned ER | 1 | 110 | 1 |
| Dendritic Segment | Initial PM† | 45 | 62 | 0 |
| | Watertight PM* | 1 | 0 | 1 |
| | Conditioned PM | 1 | 0 | 1 |

† Betti number computation in GAMer 2 only supports manifold surface meshes. Initial meshes have been curated in a best faith effort to preserve the initial topology (i.e., edges connected to three or more faces are cleaned up while holes and disconnected components are untouched)

* These meshes have been edited to be watertight. Furthermore disconnected components which are artifacts of earlier workflow steps have been reconnected.

geometries. Current studies of membrane mechanics often assume that the initial membrane configuration is flat or spherical. However membranes are rarely so well behaved and to the best of our knowledge, currently no estimates of the curvatures of the plasma membrane or internal membranes exist.

Using the conditioned surface meshes, the curvature can be estimated using methods from discrete differential geometry. In GAMer 2, we have implemented the algorithms to compute curvatures as described by Cazals and Pouget (JETS) [95, 96] and Meyer et al. (MDSB) [97]. We note that the curvature values produced by GAMer 2 are estimates suitable for qualitative comparison and visualization only. The JETS algorithm fits an osculating jet to a local patch using interpolation or approximation. From this fitted jet, the curvature is calculated. Depending on the fineness of the mesh, the jet order, and the details of patch selection, the error in the curvature may vary. On the other hand, Borelli et al. has previously showed that the estimation of the Gaussian curvature using the angle defect, which is used by the MDSB algorithm, is valid only for regular meshes with a vertex valency of 6 [98]. The robust calculation of curvature estimates from discrete meshes remains an open problem and the accuracy of many approaches are surveyed by Váša et al. in Ref. [99]

Here, we use the meshes produced by GAMer 2 to calculate the curvature of the geometries using the MDSB algorithm. The principal curvatures $\kappa_1$ and $\kappa_2$ are shown in Figs 8 and 9, respectively. We note that we have adopted the sign convention where a negative curvature corresponds to a convex region (cf. S1 Table). A comparison of the MDSB estimate and JETS is shown in S1 Fig. We find that both algorithms produce qualitatively similar results.

Looking at the first principal curvature, Fig 8, corresponding to the maximum curvature of the local region, we find that the distribution of $\kappa_1$ spans both positive and negative values, centered around zero for both the PM and ER. The negative regions of $\kappa_1$ are in regions where

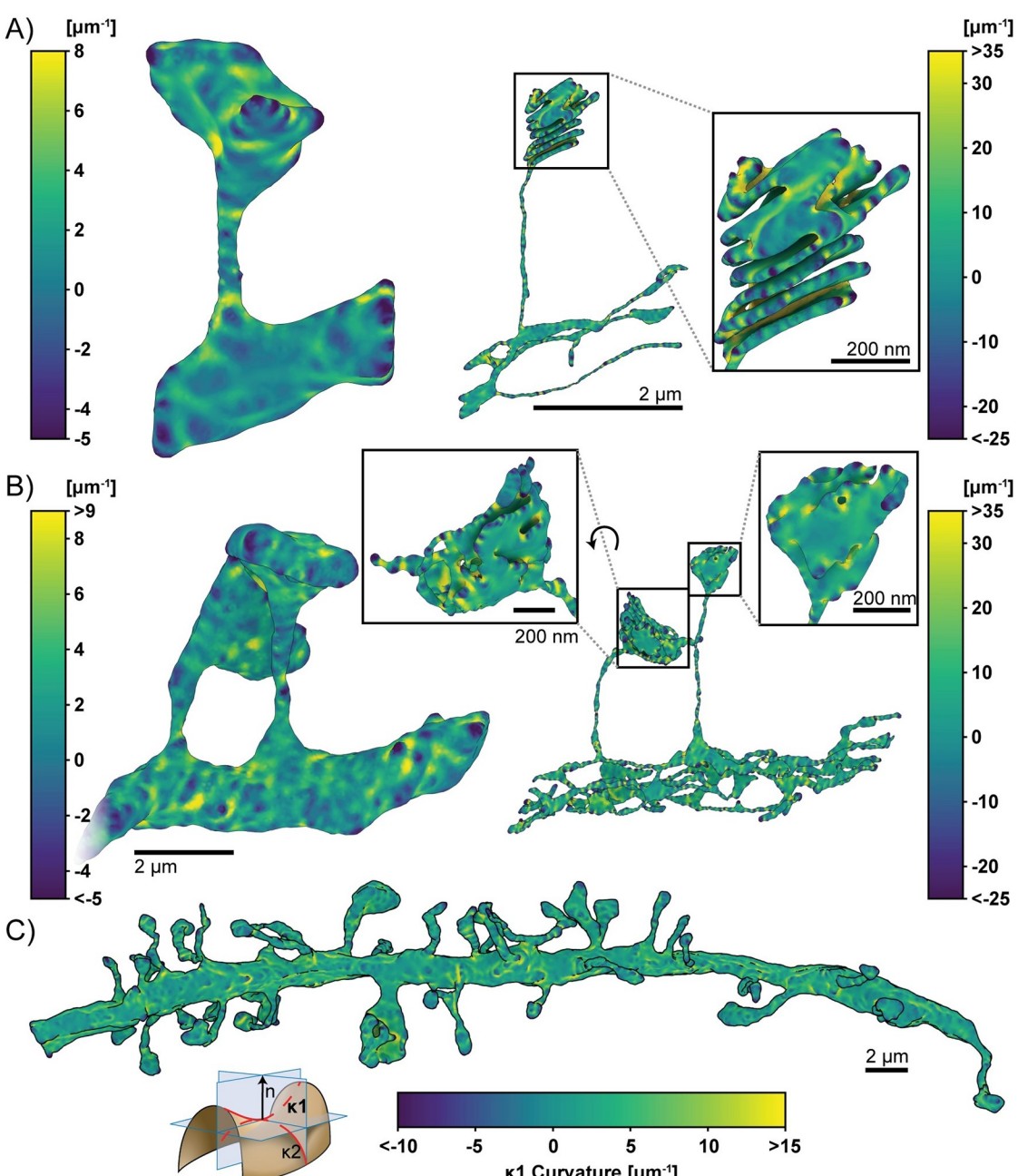

**Fig 8. Estimated first principal curvatures of the spine geometries.** The signed first principal curvature, corresponding to the maximum curvature at each mesh point, is estimated using GAMer 2. Color bars correspond to curvature values with units of $\mu m^{-1}$. We have adopted the sign convention where negative curvature values refer to convex regions. Geometries are A) single spine model, left: plasma membrane, right: endoplasmic reticulum; B) two spine model, left: plasma membrane, right: endoplasmic reticulum; and C) plasma membrane of the dendritic branch model. Scale bars: full geometries 2 $\mu$m, inlays: 200 nm. Curvature schematic modified from Wikipedia, credited to Eric Gaba (CC BY-SA 3.0).

the membrane is convex and the positive regions are in regions where the membranes are concave.

The second principal curvature, which corresponds to the minimum curvature of the local region, shows a different behavior (Fig 9). We first observe that for all geometries, this value is

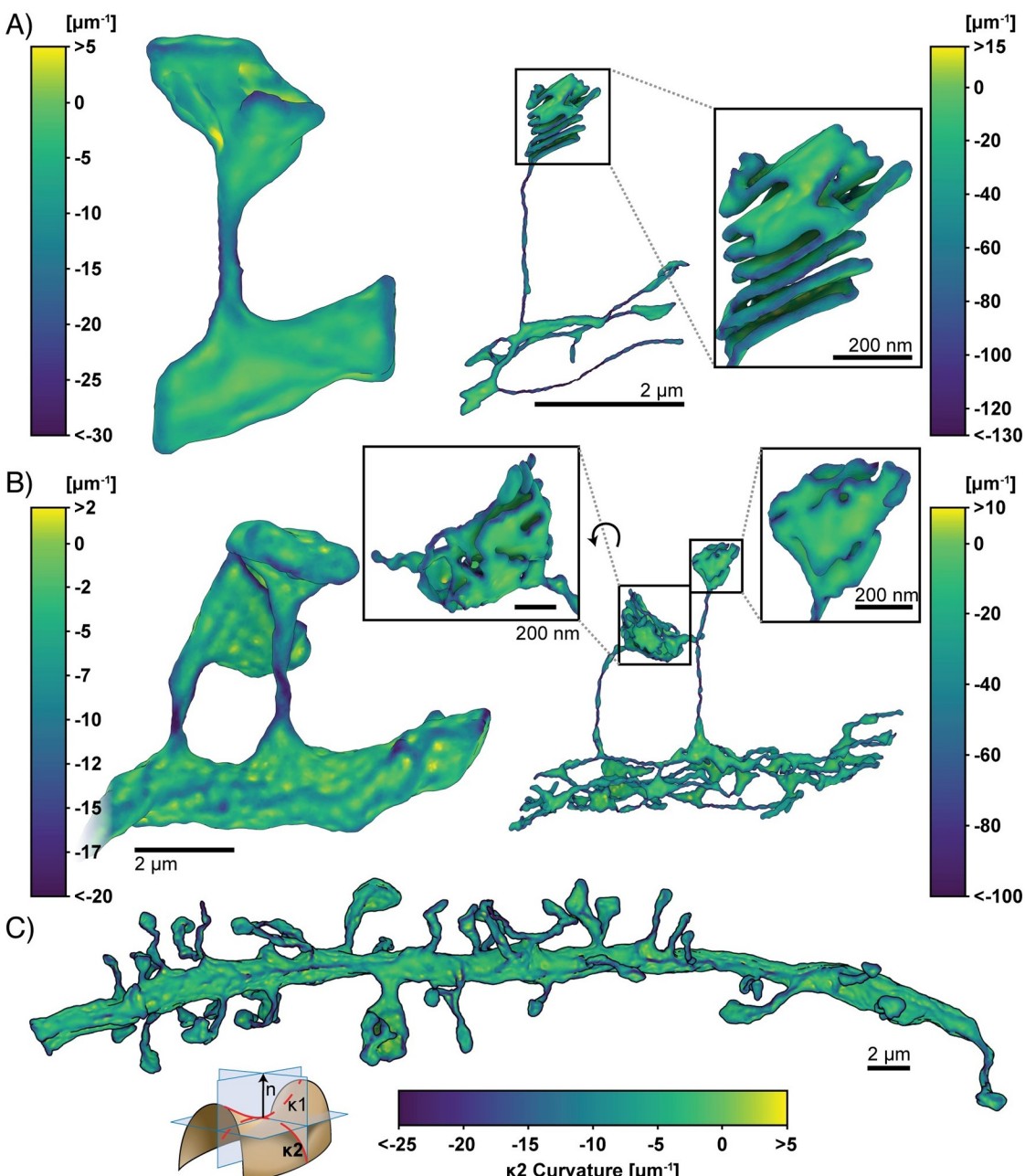

**Fig 9. Estimated second principal curvatures of the spine geometries.** The signed second principal curvature, corresponding to the minimum curvature at each mesh point, is estimated using GAMer 2. Color bars correspond to curvature values with units of $\mu$m$^{-1}$. We have adopted the sign convention where negative curvature values refer to convex regions. Geometries are A) single spine model, left: plasma membrane, right: endoplasmic reticulum; B) two spine model, left: plasma membrane, right: endoplasmic reticulum; and C) plasma membrane of the dendritic branch model. Scale bars: full geometries 2 $\mu$m, inlays: 200 nm. Curvature schematic modified from Wikipedia, credited to Eric Gaba (CC BY-SA 3.0).

primarily negative. The regions of high bending such as the folds of the spine apparatus in the spine head (Fig 9A and 9B, left panels) are highly curved and are connected by sheets with low curvature. The curvature along the entire dendrite highlights that the structure is mostly characterized by low curvature throughout with regions of concentrated high curvature (Fig 9C).

The positive regions of $\kappa_2$ are in regions where the membrane is convex and the negative regions are in regions where the membranes are concave. Thus using the meshes generated from GAMer 2, we are able to quantify the curvatures along the plasma membrane and the internal organelle membranes using tools from discrete differential geometry. In addition to estimating the curvatures, we can use the mesh models to interrogate the impacts of curvature on signaling.

## Simulations of a coupled volume and surface diffusion model

To showcase how simulations performed on meshes of realistic biological geometries can elucidate structure-function relationships, we reproduce the results of Ref. [100] on a dendritic spine. The one spine geometry was used as the spatial domain for a numerical simulation using the finite element method. Consider the reaction,

$$A + X \underset{k_{\mathrm{off}}}{\overset{k_{\mathrm{on}}}{\rightleftharpoons}} B,$$

where A is a cytosolic component which binds to X, a membrane bound component, to produce B, another membrane bound component. The governing equations consist of a volumetric Partial Differential Equation (PDE),

$$\frac{\partial A}{\partial t} = D_A \Delta A \quad \text{in} \quad \Omega, \tag{8}$$

two surface PDEs,

$$\frac{\partial X}{\partial t} = D_X \Delta_S X - k_{\mathrm{on}} A|_{\partial\Omega} X + k_{\mathrm{off}} B \quad \text{on} \quad \partial\Omega \tag{9}$$

$$\frac{\partial B}{\partial t} = D_B \Delta_S B + k_{\mathrm{on}} A|_{\partial\Omega} X - k_{\mathrm{off}} B \quad \text{on} \quad \partial\Omega, \tag{10}$$

and a boundary condition for A which couples all three species at the interface:

$$D_A(\mathbf{n} \cdot \nabla A) = -k_{\mathrm{on}} A X + k_{\mathrm{off}} B \quad \text{on} \quad \partial\Omega. \tag{11}$$

$D_A$, $D_X$, and $D_B$ are the diffusion coefficients for A, X, and B respectively. $\mathbf{n}$ is the outwardly-oriented unit normal vector, $\Delta$ is the standard Laplacian operator, $\Delta_S$ is the Laplace-Beltrami operator, $\Omega$ is the volumetric (cytosolic) domain, and $\partial\Omega$ is the surface (plasma membrane) domain (illustrated in Fig 10A). The parameters used in this system are as follows: $k_{\mathrm{on}} = 1\ \mu M^{-1} s^{-1}$ $k_{\mathrm{off}} = 0.1 s^{-1}$, $D_A = 0.1 \ldots 300\ \mu m^2 s^{-1}$, $D_X = 0.1\ \mu m^2 s^{-1}$, and $D_B = 0.01 \mu m^2 s^{-1}$. The initial conditions were set to $A(t = 0) = 1.0\ \mu M$, $X(t = 0) = 1000$ molecules $\mu m^{-2}$, and $B(t = 0) = 0$ molecules $\mu m^{-2}$. The initial concentration of X was set to a large value such that it would not be a rate-limiting factor.

Multiplying each PDE by a test function, integrating over their respective domains, and applying the divergence theorem results in the variational or weak form of the problem. After discretizing the time derivatives using the backward Euler method with time-step size $\delta t$, and decoupling the volumetric and surface PDEs using a first-order operator splitting scheme the

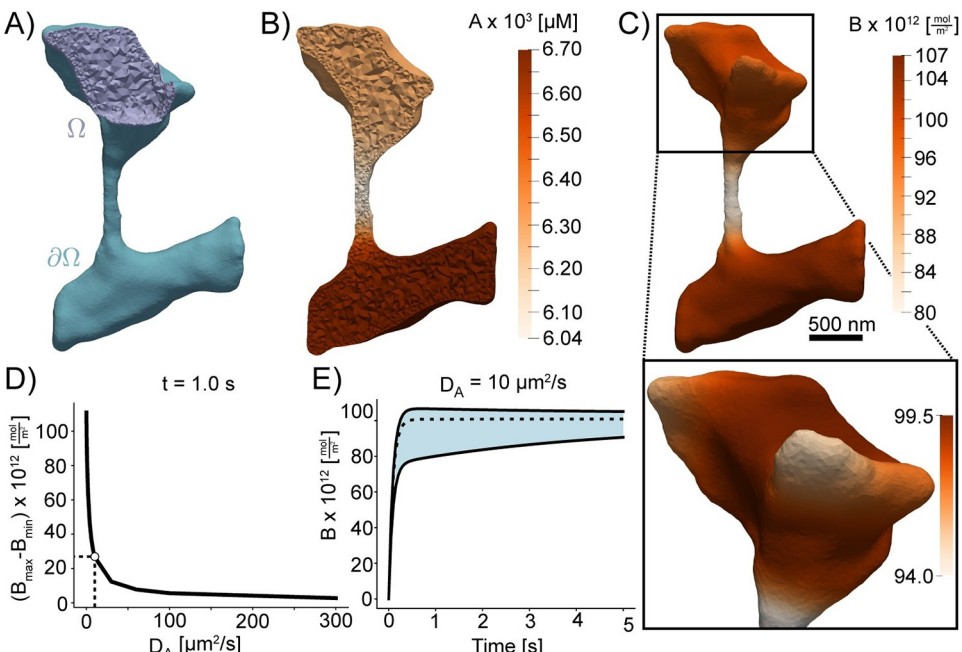

**Fig 10. Simulations of coupled surface volume diffusion.** A) Illustration of the domains for the volume and surface PDEs. B) and C) The concentrations of species A and B, respectively, at $t = 1.0s$ when $D_A$ is set to 10 μm²/s. D) Difference between maximum and minimum values of B at $t = 1.0s$; the point $D_A = 10$ μm²/s corresponding to panels B) and C) is marked. E) the minimum, mean, and maximum of B over time when $D_A = 10$ μm²/s; a vertical bar is drawn at $t = 1.0s$. Scale bar: 500nm.

system becomes:

$$\int_{\Omega} \frac{A^{(n+1)} - A^{(n)}}{\delta t} \nu_A + D_A \nabla A^{(n+1)} \cdot \nabla \nu_A \, d\Omega + \int_{\partial\Omega} k_{on} A^{(n+1)} \tilde{X} \nu_A - k_{off} \tilde{B} \nu_A \, d\Gamma = 0, \qquad (12)$$

$$\int_{\partial\Omega} \frac{X^{(n+1)} - X^{(n)}}{\delta t} \nu_X + D_X \nabla_S X^{(n+1)} \cdot \nabla_S \nu_X + k_{on} \tilde{A} X^{(n+1)} \nu_X - k_{off} B^{(n+1)} \nu_X \, d\Gamma = 0, \qquad (13)$$

$$\int_{\partial\Omega} \frac{B^{(n+1)} - B^{(n)}}{\delta t} \nu_B + D_B \nabla_S B^{(n+1)} \cdot \nabla_S \nu_B - k_{on} \tilde{A} X^{(n+1)} \nu_B + k_{off} B^{(n+1)} \nu_B \, d\Gamma = 0. \qquad (14)$$

Here, $\tilde{A}$, $\tilde{X}$, and $\tilde{B}$ represent the most recent estimates of $A^{(n+1)}$, $X^{(n+1)}$, and $B^{(n+1)}$. At each time-step Eq 12 is solved to estimate $A^{(n+1)}$, this estimate is then used in Eqs (13) and (14) to obtain an estimate for $X^{(n+1)}$ and $B^{(n+1)}$ which are used again in Eq 12 to further improve the estimate of $A^{(n+1)}$. This cycle continues until a satisfactory convergence criterion is met.

Note that Eqs (9) and (10) are PDEs that govern phenomena occurring entirely on the surface $\partial\Omega$. The geometry of the surface $\partial\Omega$ is encoded in the differential operators $\nabla_S$ and $\Delta_S$ which represent the surface gradient and Laplace-Beltrami operators, respectively cf. [14, 101]. This class of PDEs with spatial domains being represented by two-dimensional surfaces, or more generally Riemannian $n$-manifolds, are known as *geometric PDEs*, and arise in a number of areas of pure mathematics and mathematical physics, as well as in applications in science and engineering. Unfortunately, two distinct challenges arise in developing numerical methods for geometric PDEs with the necessary convergence properties to allow for drawing

scientific conclusions from simulations. The first is the necessity of treating the continuous curved spatial manifold only approximately, using some type of computable discrete proxy (such as an interpolatory triangulation of a smooth two-surface), and then accounting for the impact of this domain approximation on the overall error in a numerical simulation. The second difficulty is the need to approximate the metric of the smooth surface that appears in the definition of the Laplace-Beltrami operator itself, using some type of computable approximation (such as a polynomial), and again accounting for the impact of this second distinct approximation on the overall error in the numerical simulation of phenomena on the surface.

*Surface finite element methods* have emerged [101–104] over the last few years as an approach to developing finite element methods for this class of problems that have well-understood convergence properties, and are both efficient and reliable. The method is formulated on a "flat" triangulated approximation of the curved domain surface, and the error produced by this approximation is then controlled using a "variational crimes" framework known as the *Strang Lemmas*. Our recent work in this area leverages the Finite Element Exterior Calculus framework [105] to provide a more general error analysis framework for surface finite element methods on *n*-surfaces, for static linear and nonlinear problems [106, 107], as well as for evolution problems on surfaces [108, 109]. Surface finite element methods for geometric PDE have the advantage of allowing for the use of standard finite element software originally developed for standard (non-geometric) PDE problems in two-dimensional "flat" domains or three-dimensional volumes, after a fairly simple modification to the reference element maps commonly used by such software packages. Our approach here is to make use of the standard finite element software package FEniCS [110], and to use surface finite element modifications to FEniCS (described e.g. in [14]) for solving our geometric PDE Eqs (9) and (10) above.

To demonstrate the role of membrane shape in coupled reaction-diffusion simulations of membrane and volume components, we simulated the reaction of a volume component A reacting with membrane bound species X to form membrane bound species B. The volumetric domain, $\Omega$, and the boundary domain, $\partial\Omega$, are labeled in Fig 10A. Shown in Fig 10B and 10C, are the concentrations of species A and B in the volume and on the surface respectively. We found that the shape of the dendrite has a significant effect on the formation of B on the surface and on the depletion of A in the volume. In regions of high curvature, such as the small protrusions in the head, we found that the density of B is lower because of local depletion of A. This effect can be seen very clearly along the spine neck, where the surface area to volume ratio is high and the resulting density of B is lower than in the dendrite. To investigate if the diffusion coefficient of A affects these results, we varied the diffusion of A and analyzed its effects on the surface distribution of B. As expected, we found that as the diffusion coefficient of A is increased, the effects of local depletion are weakened (Fig 10D). Fig 10E shows the maximum and minimum concentrations of B plotted with respect to time. We find that there is a large initial difference in rates of B formation, as indicated by the large gap between the maximum and minimum concentrations, subsequently this gap narrows as A is slowly depleted from the volume. Thus, we show that the meshes generated from GAMer 2 can be used for systems biology with coupled surface-volume interactions in realistic geometries.

**Mesh convergence analysis.**   Next, we illustrate the effects of GAMer 2 mesh conditioning on FEA results for the model described above. We investigate the performance improvement as a function of rounds of conditioning. A common error metric used in FEA is the $L_2$ norm of the difference between a solution computed on a coarser mesh ($u'$) and a solution

computed on a very fine mesh, which is taken to be the ground-truth ($u$), i.e.,

$$\varepsilon_{L_2} = \left( \int_\Omega (u' - u)^2 \, d\Omega \right)^{\frac{1}{2}}. \tag{15}$$

This is a standard procedure that can be used to measure $h$-refinement convergence rates; however between iterations of GAMer 2 algorithms the boundaries of the mesh are perturbed slightly. Attempting to use $\varepsilon_{L_2}$ as an error metric is problematic as its integrand is undefined in regions where $\Omega'$, the domain of $u'$, and $\Omega$ do not intersect.

Therefore, to illustrate the convergence of solutions as the mesh quality is improved using GAMer 2, we used an error metric based on the relative difference in total molecules,

$$\varepsilon_{\mathrm{rel}} = \left| \frac{\int_{\Omega'} u' \, d\Omega' - \int_\Omega u \, d\Omega}{\int_\Omega u \, d\Omega} \right|. \tag{16}$$

Fig 11(D)–11(G) shows tetrahedral meshes generated at intermediary steps during the GAMer 2 refinement process. For each mesh, a given number of surface mesh smoothing iterations was performed; any remaining artifacts that would prevent tetrahedralization such as intersecting faces were removed and the resultant holes were re-triangulated. The surface meshes were all tetrahedralized using TetGen with the same parameters. As the surface mesh Fig 11A and 11B show how as the distribution of surface mesh angles improves, the distribution of radius-ratios in the corresponding tetrahedral mesh improves. The system Eqs (8) to (11) was solved on the aforementioned tetrahedral meshes for a single time-step and the relative error for B was computed using Eq (16). The simulation on the most refined mesh (Fig 11G) was assumed to be the ground-truth. Fig 11C shows that the relative error consistently decreased as a result of further mesh conditioning in GAMer 2. This analysis highlights not only the importance of using a high quality mesh in FEA but also that GAMer 2 can generate such high quality meshes.

## Simulation of reaction-diffusion equations on a dendrite

Using the mesh of the dendritic segment we simulated $N$-methyl-D-aspartate Receptor (NMDAR) activation due to a Back Propagating Action Potential (BPAP) and Excitatory Postsynaptic Potential (EPSP) along the entire dendrite shown in Fig 12 and S3 Movie. Because the goal of this simulation was not to show biological accuracy, but rather to demonstrate that our approach is capable of producing biophysically relevant FEA simulations, we use a simplified version of the model found in Bell et al. [111].

We model a BPAP and EPSP which stimulates NMDAR opening and calcium ion influx into the cell. The reaction-diffusion of $u$, corresponding to calcium ion concentration, is described as follows,

$$\frac{\partial u}{\partial t} = D\Delta u - \frac{u}{\tau} \quad \text{in} \ \ \Omega, \tag{17}$$

where $D$ is the diffusion coefficient of $u$, $\Delta$ is the Laplacian operator, $\tau$ is a characteristic decay time, and $\Omega$ is the volumetric domain. We define boundary conditions corresponding to the ionic flux through NMDARs, $J_{\mathrm{NMDAR}}$, lining the post synaptic density, $\partial\Omega_{\mathrm{psd}}$,

$$D(\mathbf{n} \cdot \nabla u) = J_{\mathrm{NMDAR}}(t) \quad \text{on} \ \ \partial\Omega_{\mathrm{psd}}, \tag{18}$$

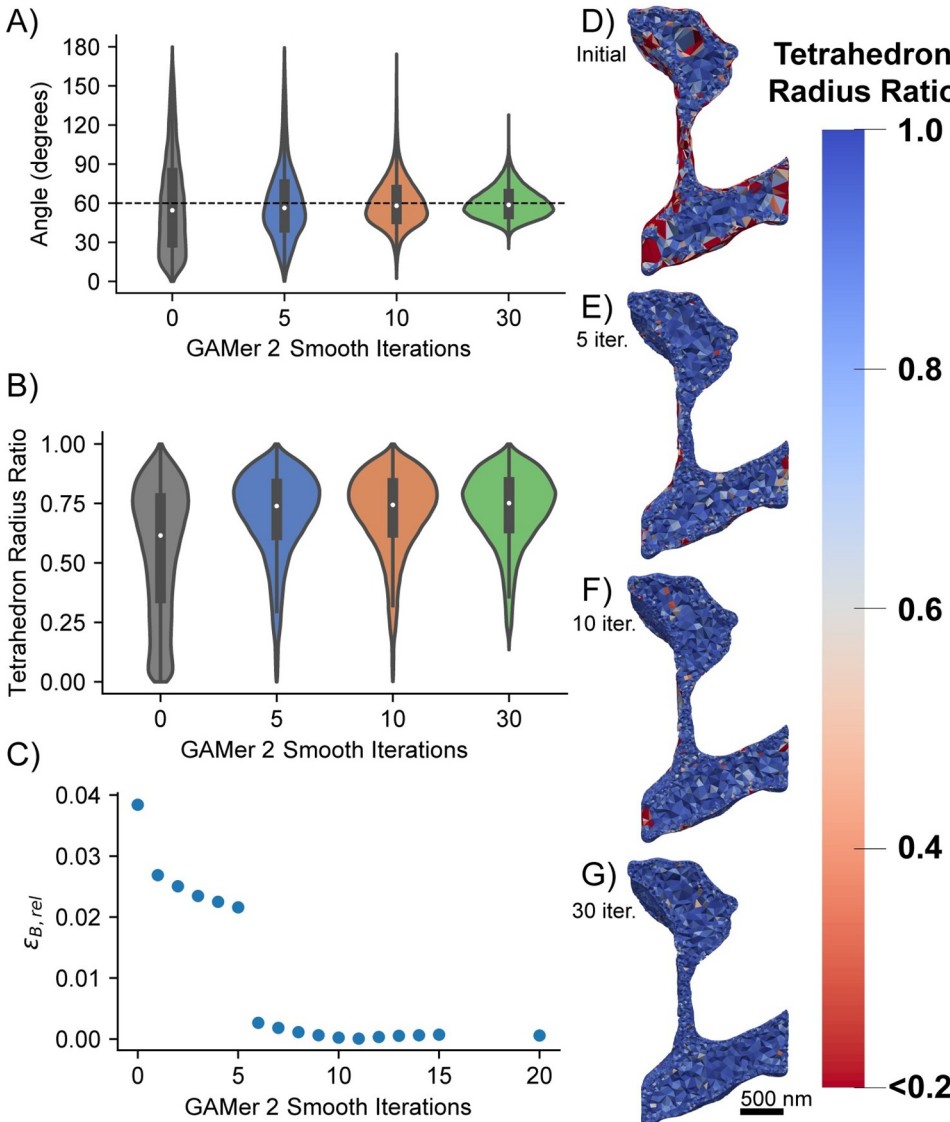

**Fig 11. GAMer 2 mesh conditioning reduces error in the simulated result.** A) Distribution of angles on the surface mesh after 0, 5, 10, 30 smoothing operations. B) Distribution of tetrahedral radius-ratios*. The tetrahedron radius-ratio is defined as $\frac{3r_i}{r_o}$ where $r_i$ is the radius of the inscribed sphere and $r_o$ is the radius of the circumsphere (a value of 1 corresponds to an equilateral tetrahedron). C) Relative error (Eq 16) of $B$ when solving Eqs (8) to (10) for a single time step. (D-G) Tetrahedral radius-ratios after 0, 5, 10, 30 smoothing iterations. Generally, for simulation using the finite element method, most radius-ratios should be greater than $\frac{1}{3}$ [90]. *Artifacts which prevented tetrahedralization, e.g. intersecting faces, were removed and the holes were remeshed at each step.

where **n** is the outwardly-oriented unit normal vector, and $J_{\mathrm{NMDAR}}$ is of the form,

$$J_{\mathrm{NMDAR}} = G_{\mathrm{NMDAR}}(t)B(V)(V(t) - V_{\mathrm{rest}})\alpha. \qquad (19)$$

$G_{\mathrm{NMDAR}}(t)$ is a variable conductance which accounts for deactivation of the receptor, $B(V)$ is a term which accounts for $Mg^{2+}$ inhibition, the voltage difference $V(t) - V_{\mathrm{rest}}$ is prescribed to emulate a BPAP and EPSP, and $\alpha$ is a scaling term which groups factors such as probability of opening, receptor area density at the PSD, etc.

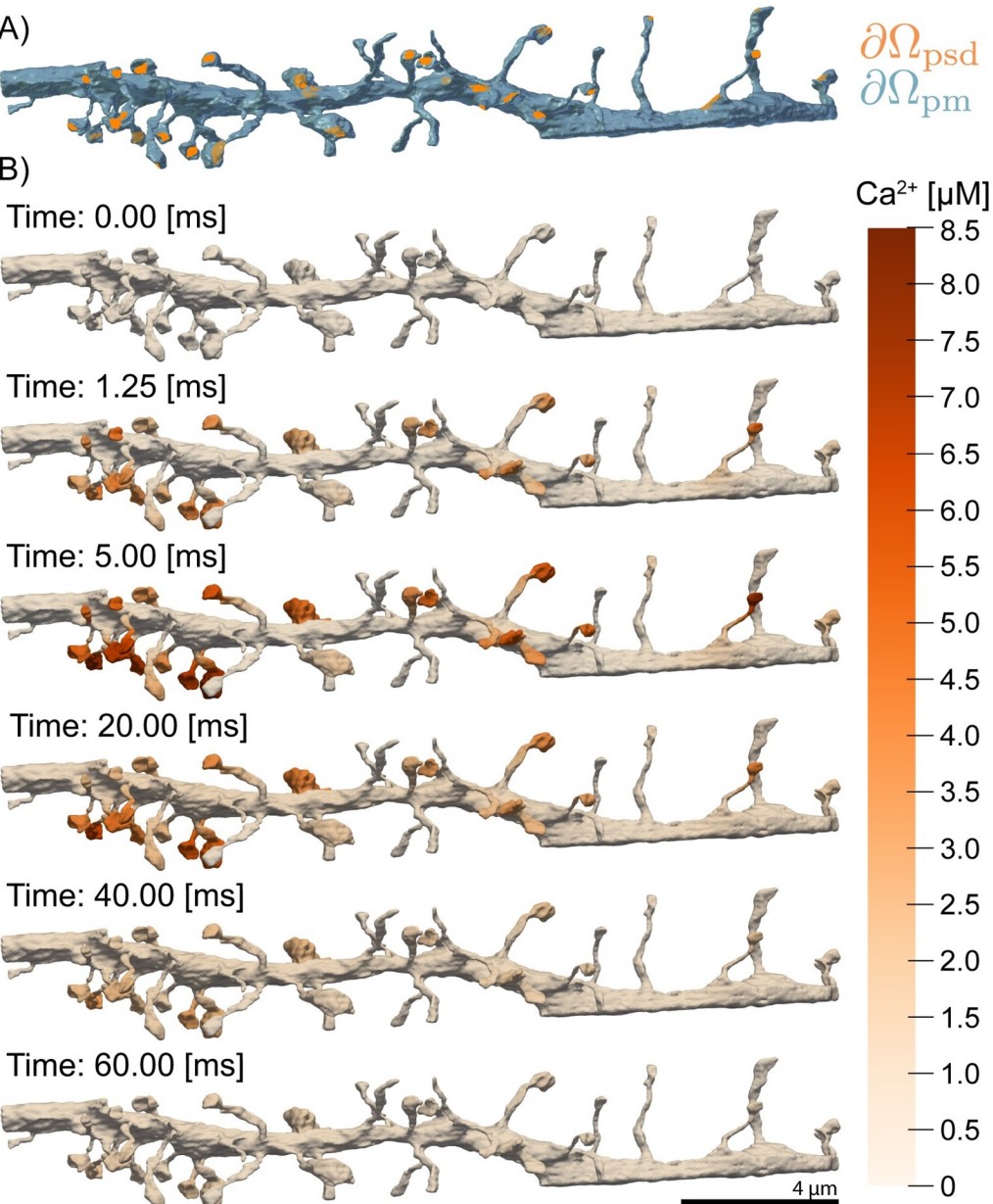

**Fig 12. Time series of calcium dynamics from *N*-methyl-D-aspartate Receptor (NMDAR) opening in response to a prescribed membrane voltage trace in a full dendritic segment.** A) Boundaries demarcating the Plasma Membrane (PM) and Postsynaptic Density (PSD) are shown in blue and orange respectively. B) Snapshots of calcium ion concentration throughout the domain are also shown for several time points. We apply a voltage corresponding to a back propagating action potential and excitatory postsynaptic potential. NMDAR localized at the PSD membrane, open in response to the voltage and calcium flows into the cell. Over time, the NMDAR close, and calcium is scavenged by calcium buffers.

On the remainder of the plasma membrane which we denote as $\partial\Omega_{pm}$, we enforce the no-flux boundary condition,

$$D(\mathbf{n} \cdot \nabla u) = 0 \quad \text{on} \quad \partial\Omega_{pm}. \tag{20}$$

At time $t = 0$, we set the initial concentration of calcium ions to naught throughout the volume of the dendrite,

$$u(\mathbf{x}, t = 0) = 0 \quad \text{in} \quad \bar{\Omega}. \tag{21}$$

Where $\bar{\Omega}$ is the union of the volumetric and boundary domains,

$$\bar{\Omega} \equiv \Omega \cup \partial\Omega. \tag{22}$$

The surface of the geometry is composed of only post synaptic density and plasma membrane,

$$\partial\Omega \equiv \partial\Omega_{\text{psd}} \cup \partial\Omega_{\text{pm}}. \tag{23}$$

Finite element simulations of this model were solved using `FEniCS` [110].

In this simplified model, we assume that the back propagating potential stimulates the entirety of the dendritic branch simultaneously, leading to the opening of NMDARs localized to the PSD and an influx of calcium ions. Several representative snapshots of Ca$^{2+}$ concentration over time, across the geometry, are shown in Fig 12. The Ca$^{2+}$ transient can be probed by monitoring the concentration at specific locations, shown in Fig 13. As expected, we first observe that the calcium dynamics are spine size, spine shape and PSD-dependent. Probes 1 and 2 in Fig 13 are in different spine heads and report differing Ca$^{2+}$ transients. Furthermore, we observe that the narrow spine necks act as a diffusion barrier to calcium, preventing diffusing calcium ions from entering the dendritic shaft as illustrated by probe 3 in Fig 13. This behavior of the spine neck as a diffusion barrier is consistent with other observations in the literature [112–115].

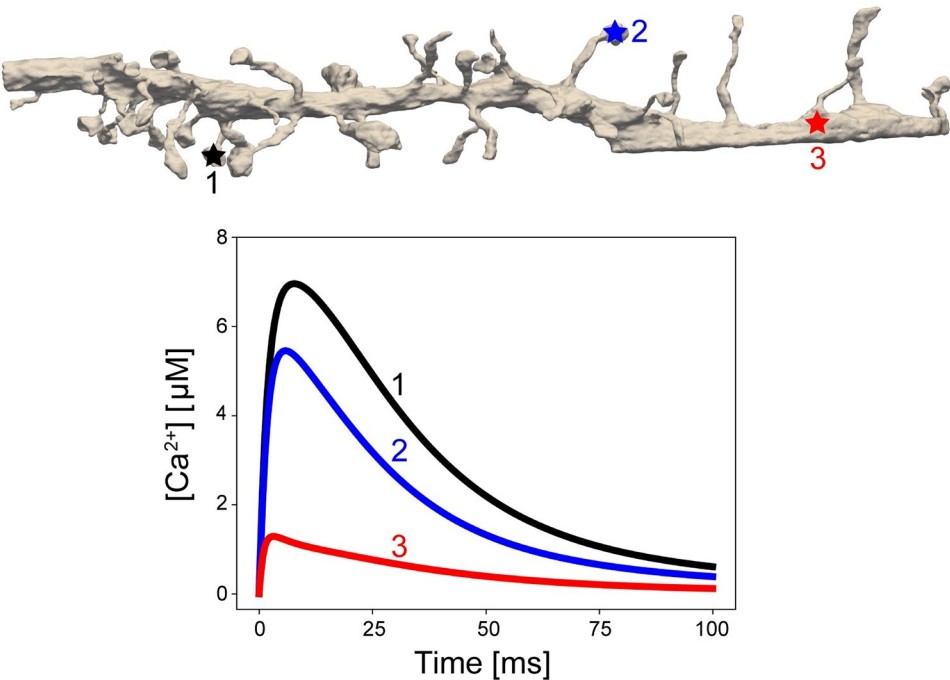

**Fig 13. Representative traces of Ca$^{2+}$ concentration over time at three positions.** Spine and PSD morphology affect the calcium ion dynamics. For traces 1 and 2, variations in the PSD area and spine head volume lead to different peak calcium ion concentrations. At point 3, the calcium ion concentration values are diminished due to both calcium buffering in the cytosol and the spine neck behaving as a diffusion barrier.

This example demonstrates that the meshes produced by GAMer 2 through the workflow are directly compatible with finite element simulations and will allow for the generation of bio-physically relevant hypotheses.

## Discussion

The relationship between cellular shape and function is being uncovered as systems, structural biology, and physical simulations converge. Beyond traditional compartmentalization, plasma membrane curvature and cellular ultrastructure have been shown to affect the diffusion and localization of molecular species in cells [100, 116]. For example, fluorescence experiments have shown that the dendritic spine necks act as a diffusion barrier to calcium ions, preventing ions from entering the dendritic shaft [112]. Complementary to this and other experiments, various physical models solving reaction-diffusion equations in idealized geometries have been developed to further interrogate the structure-function relationships [51, 100, 111, 117–119].

An important next step will be to expand the spatial realism of these models to incorporate realistic geometries as informed by volume imaging modalities. Our tool GAMer 2 serves as an important step towards filling the need for community driven tools to generate meshes from realistic biological scenes. We have demonstrated the utility of the mesh conditioning algorithms implemented in GAMer 2 for a variety of systems across several length scales and upwards of hundreds of thousands of triangles. The surface mesh conditioning algorithms in GAMer 2 are local operations, which therefore scale linearly with the number of vertices. The hardware requirements to run GAMer 2 are modest and all meshes shown in this work were processed on a laptop. In the future, GAMer 2 can be improved to support parallel (shared and/or distributed) processing and conditioning on GPUs. We refer the reader to Ref. [83] for the analysis of runtime complexities of algorithms in TetGen. The volume meshes that result from our tools are of high quality (Fig 7) and we show that they can be used for estimating membrane curvatures (Figs 8 and 9) and in finite element simulations of reaction-diffusion systems (Figs 10, 12 and 13).

Bundled with GAMer 2 we include the BlendGAMer add-on which exposes our mesh conditioning algorithms to the Blender environment. Blender acts as a user interface that provides visual feedback on the effects of GAMer 2 mesh conditioning operations. Blender also enables the painting of boundaries using its many mesh selection tools. Beyond the algorithms in GAMer 2, Blender also provides an environment for manual curation of mesh artifacts.

Current meshing methods are limited by the need for human biological insight. Experimental setups for volume electron microscopy are arduous and often messy. Microscopists take great care to optimize the experimental conditions, however small variations can lead to sample contamination, tears, precipitation of stain, or other problems. Many of these issues will manifest as artifacts on the micrographs, which makes it challenging to evaluate the ground truth. Automated segmentation algorithms using computer vision and machine learning approaches can fail as a result of these artifacts, and biologists will default to the time-tested, reliable but error-prone mode of manually tracing boundaries.

This is a unique opportunity for biological mesh generation to differentiate from other meshing tools employed in other engineering disciplines. To account for the challenges inherent to biology and wet experiments along with physical simulations, the realization of an automated mesh generation pipeline will require the development of specialized algorithms which tightly couple information across the workflow. As additional annotated datasets become

available, machine learning models can be trained to perform tasks that are currently manually executed, such as reconnecting disconnected ER tubules.

The approach and tools presented here, coupled with advances in localization of various membrane proteins [120], brings us closer to the goal of *in silico* biology within realistic geometries. Towards the goal of making 3D cell modeling more routine, experimentalists can contribute by sharing segmented datasets from their work along with biological questions of interest. In exchange, modelers can generate testable predictions and measurements inaccessible to current experimental methods. Specifically, we anticipate that models enabled using `GAMer 2` will be of significant interest to two broad communities in computational biology: membrane biophysicists, focused on the analysis and simulation of membrane shapes, curvature generation, and membrane-protein interactions, and systems biologists, focused on understanding how cell shape and internal organization can impact signal transduction and the dynamics of second messenger microdomains. Through this interdisciplinary exchange, any gaps in our current meshing workflows will be identified and patched.

## Conclusion

In this study, we present our mesh generation code `GAMer 2` and described several applications going from contours of electron micrographs as input to generate surface and volume meshes that are compatible with finite element simulations for reaction-diffusion processes. Using the resulting meshes, we have demonstrated the spatio-temporal dynamics of calcium influx in multiple spines along a dendrite. Future efforts will focus on the development of biologically relevant models and generation of experimentally testable hypotheses.

## Supporting information

**S1 Fig. Comparison of Meyer-Desbrun-Schröder-Barr (MDSB) and Cazals-Pouget (JETS) curvature estimates for the single spine geometry.**
(PDF)

**S1 Table. Local geometry associated with signs of principal curvatures according to our sign convention.**
(PDF)

**S1 Appendix. Protocols for the generation and comparison of meshes between software.**
(PDF)

**S1 Movie. Animation proofing the initial mesh against the segmented micrographs.** The PM (blue) is rendered as a wireframe. The ER (yellow) is displayed as a solid surface. Purple patches in the segmentation correspond to regions where the PSD is localized. It is using these labeled patches that the boundary marking is generated.
(MP4)

**S2 Movie. Animation proofing the `GAMer 2` conditioned mesh against the segmented micrographs.** The PM (blue) is rendered as a wireframe. The ER (yellow) is displayed as a solid surface. Purple patches in the segmentation correspond to regions where the PSD is localized.
(MP4)

**S3 Movie. Trajectory of calcium ion signaling in the full dendrite geometry.**
(MP4)

## Acknowledgments

We would like to thank Prof. Pietro De Camilli and coworkers for sharing their datasets from Wu et al. [7]. We also thank Dr. Matthias Haberl, Mr. Evan Campbell, Profs. Brenda Bloodgood and Mark Ellisman for helpful discussion and suggestions. CTL especially thanks Dr. John B. Moody, and Mr. Mason V. Holst for discussions on GAMer 2 code development along with Dr. Tom Bartol for additional help with using Blender and the design of Blender add-ons. We thank Ms. Miriam Bell, Ms. Kiersten Scott, Ms. Jennifer Fromm, and Dr. Donya Ohadi for critical comments and suggestions for improving this manuscript.

## Author Contributions

**Conceptualization:** Christopher T. Lee, Rommie E. Amaro, J. Andrew McCammon, Michael Holst, Padmini Rangamani.

**Data curation:** Christopher T. Lee, Justin G. Laughlin, Nils Angliviel de La Beaumelle.

**Formal analysis:** Christopher T. Lee, Justin G. Laughlin, Padmini Rangamani.

**Funding acquisition:** Rommie E. Amaro, J. Andrew McCammon, Michael Holst, Padmini Rangamani.

**Investigation:** Christopher T. Lee, Justin G. Laughlin.

**Methodology:** Christopher T. Lee, Justin G. Laughlin, Michael Holst.

**Project administration:** Christopher T. Lee, Padmini Rangamani.

**Resources:** Rommie E. Amaro, J. Andrew McCammon, Michael Holst, Padmini Rangamani.

**Software:** Christopher T. Lee, Justin G. Laughlin.

**Supervision:** Rommie E. Amaro, J. Andrew McCammon, Ravi Ramamoorthi, Michael Holst, Padmini Rangamani.

**Validation:** Christopher T. Lee, Justin G. Laughlin.

**Visualization:** Christopher T. Lee, Justin G. Laughlin, Nils Angliviel de La Beaumelle.

**Writing – original draft:** Christopher T. Lee, Justin G. Laughlin, Padmini Rangamani.

**Writing – review & editing:** Christopher T. Lee, Justin G. Laughlin, Rommie E. Amaro, J. Andrew McCammon, Ravi Ramamoorthi, Michael Holst, Padmini Rangamani.

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
