## [Decision Letter · Decision Letter 0]

1 Oct 2019

Dear Dr Rangamani,

Thank you very much for submitting your manuscript 'GAMer 2: A system for 3D mesh processing of cellular electron micrographs' for review by PLOS Computational Biology. Your manuscript has been fully evaluated by the PLOS Computational Biology editorial team and in this case also by four independent peer reviewers. The reviewers appreciated the attention to an important problem, but raised some substantial concerns about the manuscript as it currently stands. While your manuscript cannot be accepted in its present form, we are willing to consider a revised version in which the issues raised by the reviewers have been adequately addressed. We cannot, of course, promise publication at that time.

While revising your manuscript, please take care to reply the points raised by the reviewers, in particular reviewers #2 and #4 who demand, among others, more thorough comparison of your results with state-of-the art methods at each individual steps of the pipeline (especially meshing algorithms). 

Sincerely,

Hugues Berry

Associate Editor

PLOS Computational Biology

Mona Singh

Methods Editor

PLOS Computational Biology

[LINK]

Reviewer's Responses to Questions

**Comments to the Authors:**

Reviewer #1: The paper from Lee et al. explores specifically one problem, the generation of meshes from segmented EM micrographs, which is usually good enough for visualization using standard algorithms, but needs to have certain constraints if scientists want to use them for simulations or morphological analysis. Authors state to have developed a full pipeline bringing from 3DEM to volume meshes including GAMer2, a mesh processing software.

Although the problem tackled is of high importance in the field, some points came directly to my attention during the reading of the manuscript.

i) Figure 1 shows the main steps of their pipeline. But reading the manuscript further, it is not entirely clear to me whether they first generate the meshes from segmentation software, and then use GAMer2 to improve them, or they rather use the segmentation itself to generate the mesh. I think this is important to clarify. It indeed rather seems that authors are first generating the meshes and then fixing it, staying to the Mesh Generation Pipeline (line 226). Authors seems to have used IMOD in their test case, which generates mesh starting from contours of segmented micrographs, while most users from the 3DEM community rather generate mesh from the masks extracted from tools such as TrakEM2.

ii) The authors state at the very beginning out loud to propose a whole pipeline, from image segmentation to simulation. Actually, the whole point of the paper, is about the remeshing, and how producing healthy meshes is critical for simualtions and analysis. This is really good, but authors might want to reconsider statements in the abstract and introduction where they say that this paper provides a whole pipeline for image processing. They do produce meshes in several ways, from different available image segmentation software, that they do not describe, therefore these should not be considered part of the method.

iii) Lines 1 to 18 are completely devoid of literature citations. Authors could cite the review from Knott and Genoud "Is EM dead?", the pioneering work from Denk and Horstmann, PLoS Biology, 2004 showing the first 3View setup, on top of my head. Also, they should cite and make some examples of few works showing how 3DEM and image segmentation have helped to visualize the three-dimensional structure of entire cells, and their content. For instance the Cell paper from Kasthuri et al., 2015, or the PLoS one paper from Calì et al., 2018, where authors shows dense reconstructions of neuropil, including mitochondria and synaptic vesicles. I think all these examples helps to contextualize their work, and authors should make a careful literature review and cite properly the sources.

iv) Blender sculpt mode is mentioned. This is just a curiosity, but does authors use sculpt to fix the meshes? It is dangerous as a smoothing tool, given that arbitrary modifications of the surface might lead to change in SA/Vol ratios, and most likely change the Surface Area of the original mesh. Authors should discuss carefully this point. This also leads to the question, whether authors are also considering to test their meshes for preserving their volume and more importantly surface area, which is sensitive to mesh manipulations.

v) going through the methods, which are convincing and very well documented, I think one very important thing to include is whether you can proofread the morphology of the mesh by superimposing the mesh itself with the original image stack. Although qualitative, it is an important thing to check, and would be good to incorporate in a figure.

vi) line 239, authors state that quantities like surface areas, volumes and curvatures can be estimated, and then correctly document how to estimate curvature, which is of interest for biophysicists. Nevertheless, surface area and volume are quantities of interest as well, especially for biologists. In both cases, I wonder whether authors accounts at all for renormalization in the order of magnitude of the image stack (nanometers or micrometers), or the simply keep measurements as arbitrary units. This is really important, as simulations then use constant and variable that are bind to proper physical units.

Finally, I am fully convinced of the added value of the meshing tool that the authors provide, and how the meshes are proper and suitable for physical simulations; nevertheless, I am far less convinced (because I don't think this is properly documented) of the degree of fidelity of the meshes compared to the real morphology, also given that authors seems to fix artefacts manually using blender tools, without proofreading the meshes together with the original image stack. I think authors should carefully address this point in particular.

Reviewer #2: General Comment

This manuscript introduces GAMer 2, a mesh processing tool which takes surface meshes reconstructed from EM images as input, and produces tetrahedral meshes that are suitable for computational simulations. This manuscript further describes a pipeline solution from EM imaging of cellular structures to simulations with meshes of these structures, by utilizing GAMer 2 as well as other third-party tools such as IMOD, Blender, TetGen, and FEniCS.

Unfortunately, the title, abstract, and some texts in this manuscript are inaccurate and misleading. GAMer 2, as at the current stage, does not directly operate on cellular electron micrographs as described in the title, but on the surface meshes generated from third-party tools like IMOD. This manuscript does not demonstrate sufficient contribution to the image acquisition, segmentation, and the initial surface mesh generation steps of the described pipeline (A to C of Fig 1). The third-party software packages used in the pipeline steps are some of the common choices for the procedures, and the usages described in the manuscript seem to be generic. Furthermore, similar workflow from contour tracing data to simulation-ready tetrahedral mesh has been introduced in at least one of the referenced publications ([10] Edwards and etc., 2014) where all steps of the pipeline were integrated into a single application. With comparison, which should have been done in the manuscript, the claim of “develop an end-to-end pipeline for this task” seems to be overstretched and should be rephrased. It is acceptable that a manuscript presents the functionalities of the software and demonstrates its application by presenting a pipeline scheme as an example. However, the contribution of the presented software and any novelty in the pipeline steps should be stated without ambiguity. The author should avoid emphasizing general usage of a third-party software package in a pipeline step as novel development, particularly when the step is not performed by the author, or the third-party software has not been integrated into the author’s software.

Despite the inaccuracy of the description, the software presented in this manuscript does contribute to the computational biology community by bridging the gap between surface mesh reconstruction of cellular structures via EM imaging, and simulation-ready mesh generation. Surface meshes generated from EM image tracing and segmentation applications are commonly for visualization only, thus inadequate for the use of computational simulation due to problems such as manifoldness. Progress has been made in recent years as referenced in the manuscript, but there remains no widely accepted, efficient solution for the problem. The software presented here can be considered as a novel candidate. The pipeline example in the manuscript also demonstrates the application of this software for common modeling interests in the field, providing workflow guidance for future research.

Detailed Comments

Title

see General Comment.

Abstract

“we develop an end-to-end pipeline for this task”

see general comment.

“We apply this pipeline to a series of electron micrographs of neuronal dendrite morphology”

According to Line 227, the starting point of the pipeline should be the contour reconstruction of the segmented micrographs rather than the micrographs themselves.

Author summary

“Here, we describe a pipeline that takes images from electron microscopy as input ...”

See comment above.

Introduction

The introduction section needs to be reorganized. At the moment, its focus switches back and forth between GAMer2 and the pipeline, mixing actual contribution from the author with third-party supports. A reasonable structure would be first to present the pipeline as a potential solution from EM imaging to computational simulation, then provide a general review on the state-of-the-art tools and solutions for each step, highlight the challenges faced in practice, finish with stating the novel contribution of this work to this pipeline, either in software development or modeling.

Line 72: “as shown in Fig. 2.” It is better to move Figure 2 to the “Meshing segments of increasing length scales” section so that the readers do not need to switch between Figure 2 and Figure 5 constantly. As here is a general introduction of mesh artifacts, referencing Figure 2 is not necessary — the same for line 76.

Methods

The author should clarify how GAMer2 differs from its previously published predecessor in terms of mesh processing algorithms, as now there seems to be a mix between GAMer and GAMer2 functionalities. If there are new contributions or significantly different from previous designs, the author should provide a clear statement here. Also, if the author decides to use GAMer as the simplification of GAMer2 in the rest of the manuscript, there should be a clear statement.

Line 109: “Finite elements simulations” -> “Finite element simulations” “the quality of the mesh” -> “mesh quality”

Line 111: “triangulation with high aspect ratios...” -> “triangles with high aspect ratios...”

Line 114: “This scheme is an extension ...” This sentence is difficult to read. The author should try to rephrase it.

Line 138: “These images can produce meshes that ... making it particularly suitable for cellular images.” This is not very convictive that the quantity of mesh elements is the cause of the failure of global optimization. As commented above, this work does not directly operate on the cellular images but their post-processed products, whose quality relies on the solutions of previous steps. One may argue that the suitability of the smoothing algorithms is simply the consequence of tool selections at previous steps, rather than the nature of the images.

Line 156: What is the reason for putting the decimation section between two smoothing algorithms?

Line 165: “finite elements simulations” -> “finite element simulations”

Line 168: “topology changing” -> “topology-changing”

Line 221: The author should provide a figure to demonstrate the usage of BlenderGAMer for boundary marking.

Line 226: “Mesh Generation Pipeline” This paragraph relates to a specific example instead of the general method, thus should be put in Results instead.

Line 254: The author needs to clarify if the modeling work presented in the manuscript provides a novel solution to the problem, or is a reproduction of results for the validation of conditioned meshes processed by GAMer2. If it is the first case, then the work presented here establishes a pipeline from mesh processing and boundary marking to modeling with techniques specific to the conditioned meshes. If it is the second case, however, this work should not be regarded as pipeline development, but an application demonstration of the processed meshes. And it should be put to the Result.

Line 323: “NMDAR” and “NMDA receptor” are both used in the manuscript, and should be unified.

Line 352: “We demonstrate the application of our pipeline to dendritic spine reconstructions of different sizes.” See general comment.

Line 396: “the distribution of the angles of the surface mesh” -> “the distribution of the triangular angles of the surface mesh”

Line 405: “but is easily and automatically resolved” -> “but is easy to be resolved automatically”

Line 398: “Prior to conditioning, the angle distribution is spread out and contains many large and small angles” A quantitative result is preferred. The author can present the means and standard deviations of both distributions for better comparison.

Line 407: “the angles of the mesh” -> “the angles of triangles of the mesh”

Line 409: “it may be necessary to increase the number of triangles to accurately capture the fine details.” All geometry details exist in the initial mesh reconstruction, so this statement is not accurate. -> “it may be necessary to increase the number of triangles to accurately capture the fine details with high mesh quality.”

Line 434: “and the PSD is marked with GAMer for use denoting a boundary condition.” GAMer or BlenderGAMer?

Line 444: “using methods from discrete differential geometry.” The author can refer to previous Method section.

Line 444: “Shown in Figs. 6 and 7 are the principal curvatures κ1 and κ2 respectively” -> “The principal curvatures κ1 and κ2 are shown in Fig 6 and 7, respectively.”

Line 449: “We also observe that tubular regions such as the neck of the spine have near uniform curvature.” The curvature values of the neck region are truncated in the colormap, so this conclusion can not be made from the current result, the same for the ER neck result. The author should analyze the values of the spines in the dendrite segment example to form a population result.

The author can also calculate the principal curvatures of the initial meshes or the conditioned meshes with different smoothing and decimation thresholds, and compare their results with the current ones to identify if these characteristics are consistent features of the cellular structures or the artifacts from the meshing process.

Line 464: “Simulations of Surface Diffusion” This is different from the “Coupled volume and surface diffusion model” simulation presented before. Can the author clarify it, please?

Line 466: “we simulated the reaction of a volume component A reacting with membrane bound species X to form membrane bound species B.” This should refer to previous sections of the model rather than repeating again.

Line 468: Why Figure 8 appears after Figure 9 in the main text?

Line 485: “GAMer 2” the name and version of the software should be consistent. This section should be put after the model examples. The author may also explore the scenario where the surface mesh remains the same while the tetrahedral mesh is progressively refined by TetGen, and compare the result.

Line 508: A reference to the previous model section is required. Alternatively, the previous section can be relocated here.

Line 543: “...that result from our tools are high quality” -> “...that result from our tools are of high quality”

Line 544: “in finite elements simulations” -> “in finite element simulations”

Line 574 ~ Line 579: This sentence requires to be rewritten.

The Conclusion

GAMer 2 should be mentioned in the conclusion, as it is the main contribution.

Figures

Fig. 1

This figure needs a complete rework as commented in the General Comment.

Fig. 2

As mentioned above, it may be better to relocation this figure.

“Left, due to the fine ultrastructure of the process, portions of the ER are not resolved by the microscope and become erroneously disconnected as separate meshes.” Alternatively, this could be caused by human error during the contour tracing process and/or insufficient triangulation solution used for surface mesh generation. The conclusion can not be made without presenting the original images as well as the contour tracing data.

Fig. 4:

“weighted and unweighted meshes” according to the figure legend as well as the order of previous subfigures, “weighted” and “unweighted” should be switched.

Fig. 5:

As mentioned above, more quantitative results are required in the explanation text such as the means and standard deviations.

Fig. 6:

"B) two spine moddel, left" -> "B) two spine model, left"

Fig. 11:

A summary of how the morphology affects the calcium dynamics is needed.

The inconsistency of font sizes and figure styles (in particular, Fig, 4C, 8F and 11) makes the read of figures slightly uncomfortable.

Reviewer #3: The manuscript describes a system of algorithms and software implementations of 3D mesh processing and then applies it to generating high-quality meshes from cellular electron micrographs. Although the algorithms are mostly taken from previous work, the present work is timely and valuable for its novel implementation of the methods and public availability of source code. The manuscript is very nicely written and the results are properly shown and discussed. I would suggest a minor change in the result section:

The authors show how mesh quality is improved after mesh processing (specifically, mesh smoothing). However, the ultimate goal of mesh processing in the current work is to provide high-quality tetrahedral meshes used for finite element analysis. For that reason, I would suggest the authors conduct some quality analysis of the tetrahedral meshes as well. For instance, one may see the tetrahedral meshes get improved while the surface meshes are being smoothed.

Reviewer #4: # OVERVIEW

Starting from electron microscopy micrographs, thiis paper presents a

pipeline reconstructing 3D meshes amenable to Finite Elements

simulations. Developing such a pipeline is an important and timely

endeavor, as the tool provided should foster our understanding of the

structure to function relationship, at length scales ranging from

micro to nanometers.

While the paper is of interest, I was not convinced for three reasons:

-- first, the specific difficulties dealt with are not carefully

specified,

-- second, the individual steps undertaken do not use state of the art

methods,

-- third, the assessment is not careful enough in terms of base-line.

The software development undertaken is very significant, and it could

be that the pipeline proposed is if very significant interest for

biologists. Yet, this has to be documented more carefully.

# MAJOR POINTS

As detailed below, I see two major problems with this paper.

## Ad hoc methods.

Starting from raw slices of electron microscopy data, the proposed

method involves the following steps: contour segmentation, primitive

mesh reconstruction from stacked contours, meshing and smoothing.

The problem is that for each of this step: (i) the problem tackled is

not formally defined -- the type of guarantee sought is not

documented, (ii) previous work is not adequately discussed, and (iii),

in several cases, the methods used are not state of the art.

Thus, overall, the proposed method is more of an ad hoc pipeline,

rather than a sequence of carefully optimized steps.

## Significance of the methods proposed for the problems at hand.

In several places, the authors discuss the intrinsic difficulties of

the problems tackled, for instance

"However, in subcellular scenes where the geometry may be tortuous and

local receptor clusters can be arbitrarily distributed on the

manifold, boundary definition is a non-trivial challenge."

"As noted earlier, finite elements simulations are sensitive to angles of the mesh. "

"In addition to the generation of FEA compatible meshes of realistic cell geometries..."

Unfortunately, this specification is rather vague. I would urge the

authors to focus on selected challenges such as "volume conservation

of organelles", "preservation of thin parts in the reconstruction /

meshing", "enforcement of boundary surfaces via constrained/conforming

Delaunay triangulations", "enforcement of bounded principal curvatures

on boundary surfaces", etc, so as to stress the added value of the

proposed pipeline.

Without a clear identification of such challenges, making strong

claims is hardly possible.

# DETAILS

* page 4. About "However, to the best of our knowledge, there is no

current free and open-source system for going from EM images to

high-quality 3D meshes, which are essential for developing reliable

high-resolution finite element simulations of cellular processes."

If one takes for granted the contours computed from the segmented

micrographs, there exists very high quality 3D meshing packages, in

particular the one from CGAL: see

https://doc.cgal.org/latest/Mesh_3/index.html

* page 4. Surface reconstruction from cross sections. Many provably

correct reconstruction methods from cross-sections have been

developed, starting with the seminal work exposed in

Boissonnat, Jean-Daniel, and Bernhard Geiger. "Three-dimensional

reconstruction of complex shapes based on the Delaunay triangulation."

Biomedical Image Processing and Biomedical

Visualization. Vol. 1905. International Society for Optics and

Photonics, 1993.

The following paper presents a mini review of recent work on this topic:

Bermano, Amit, Amir Vaxman, and Craig Gotsman. "Online reconstruction

of 3D objects from arbitrary cross-sections." ACM Transactions on

Graphics (TOG) 30.5 (2011): 113.

* page 5. About "In order to enable modeling using the shapes

represented by the contours, geometric meshes compatible with

numerical methods can be constructed."

please clarify the methods you have in mind, as specific methods

typically impose different requirements on the meshes.

* page 6. About "We note that although there exist advanced

tetrahedral mesh generation tools, such as TetWild [11], which can

generate Finite Element Analysis (FEA) compatible volume meshes

automatically from these poor quality initial surfaces, many mesh

defects are the result of the limited resolving powers of EM (e.g.,

Fig. 2A1, A2) and require more careful curation."

This is a typically example where you should be more precise to

specify the defects: are you talking about topological defects

(presence of gaps - cracks, voids), or geometric defects (typically

poorly shaped simplices such as triangles with obtuse angles or tets

such as slivers)?

* page 6. About your meshes. you should describe more carefully the

type of mesh you wish to build in order to accommodate the boundary

surfaces stemming from the segmentation steps. in particular high

quality meshes are typically related to the Delaunay triangulation,

which is not mentioned once in your main text.

* page 7. About the estimation of normal vectors ie "Vertex normals

are defined as the weighted average of incident face normals. "

It is well known that the estimation of normals and principal

curvatures should be undertaken in one step -- and not two as yo do,

as proved in the following paper

Cazals, Frédéric, and Marc Pouget. "Estimating differential quantities

using polynomial fitting of osculating jets." Computer Aided Geometric

Design 22.2 (2005): 121-146.

The corresponding software is available in CGAL, see

https://doc.cgal.org/latest/Jet_fitting_3/index.html

* page 8. About mesh smoothing. For recent advances in mesh smoothing,

see the following paper, which addresses 3 challenges at once namely

(i) geometry preservation (ii) coarsening and (iii) removal of badly

shaped elements:

Hu, Kaimo, Dong-Ming Yan, David Bommes, Pierre Alliez, and Bedrich

Benes. "Error-bounded and feature preserving surface remeshing with

minimal angle improvement." IEEE transactions on visualization and

computer graphics 23.12 (2016): 2560-2573.

* page 12. About boundaries i.e. "However, in subcellular scenes

where the geometry may be tortuous and local receptor clusters can be

arbitrarily distributed on the manifold, boundary definition is a

non-trivial challenge."

The classical methods to impose pre-defined boundaries consist of

using constrained and/or conforming Delaunay triangulations, see

again the aforementioned CGAL packages. please comment the output of

your method wrt these classical ones.

* page 14. Estimation the Gauss curvature from the angle defect. If

you assume that there exists an underlying smooth surface, the

angular defect does not yield a correct estimate of G, see

Borrelli, Vincent, Frédéric Cazals, and J-M. Morvan. "On the angular

defect of triangulations and the pointwise approximation of

curvatures." Computer Aided Geometric Design 20.6 (2003): 319-341.

* Experiments: meshes and their . The focus on the improvement of

triangle angles is great, but to fully assess the significance of the

comparison, the following critical things are missing:

- which kind of mesh does IMOD produce?

- what is the distribution of angles obtained using state of the

art methods such as those provided in CGAL and/or the the

aforementioned paper by Benes et al?

* Experiments: meshes and their topology. Are the meshed you produce

watertight? could you report Betti numbers, in order to make sure that

the topology of the regions being meshes is correct / preserved?

**Have all data underlying the figures and results presented in the manuscript been provided?**

Reviewer #1: Yes

Reviewer #2: No: Currently only the source code of GAMer 2 is provided but not the EM images, contour reconstructions and meshes presented in the examples. If they can not be provided due to the ownership, a statement must be included in the Data Availability Statement according to the data availability policy.

Reviewer #3: Yes

Reviewer #4: Yes

PLOS authors have the option to publish the peer review history of their article (what does this mean?). If published, this will include your full peer review and any attached files.

Reviewer #1: No

Reviewer #2: No

Reviewer #3: No

Reviewer #4: No

---

## [Decision Letter · Decision Letter 1]

15 Feb 2020

Dear Dr. Rangamani,

Thank you very much for submitting your manuscript "3D mesh processing using GAMer 2 to enable reaction-diffusion simulations in realistic cellular geometries" for consideration at PLOS Computational Biology. As with all papers reviewed by the journal, your manuscript was reviewed by members of the editorial board and by several independent reviewers. The reviewers appreciated the attention to an important topic. Based on the reviews, we are likely to accept this manuscript for publication, providing that you modify the manuscript according to the review recommendations.

Sincerely,

Hugues Berry

Associate Editor

PLOS Computational Biology

Mona Singh

Methods Editor

PLOS Computational Biology

[LINK]

Reviewer's Responses to Questions

**Comments to the Authors:**

Reviewer #1: I think authors have done a great job, and they certainly have carefully addressed the points I raised. I still have a couple of points that I would like to be clarified:

1 - would be useful to have a screenshot of the application running on blender. Regarding this point, they should also verify whether it runs on the latest version of Blenders which has changed substantially.

2 - would be useful to know which kind of computational power is needed to run gamer2. I think it depends of the number of vertices the tool has to process. In particular, does the processing time increase linearly, or exponentially? A basic description of the hardware needed (laptop, desktop, GPU) and a small description of the expected performaces would be useful.

Reviewer #2: The manuscript has been greatly improved. I have some comments regarding the Blender plugin and Fig1, but otherwise I am satisfied with this revision.

Fig1. "GAMer" between subfigure c and d should be "GAMer 2". This pipeline can be described in a more generic way to highlight the software's flexibility, since GAMer 2 doesn't directly couple with either IMOD or FEniCS. It may be more interesting to general readers if the supported input/output formats are given here.

L261 BlenderGAMer section: My main concern with this plugin is that Blender can only deal with surface mesh. According to my test with the current implementation of GAMer2, while the boundary tags are transferred from the boundary surface mesh to the corresponding triangles in the tetrahedral mesh, they are not propagated to the tetrahedrons. This may be sufficient for FEM simulations, but is less ideal for simulations that require the categorization of subregion tetrahedrons. I suggest the author briefly discuss the possible solution either here or the in discussion, and improve the implementation in the future.

Reviewer #4: # Overview

I am personally quite happy with the revision. From a geometric modeling perspective, the revised version indeed

* presents in many places a more accurate and specific description of the biophysical problems,

* contains a convincing evaluation of meshing algorithms (see one point below, though),

* contains a discussion of the difficulties inherent to curvature calculations in the discrete setting,

* incorporates a calculation of Betti numbers (see one point below, though).

There are a number of technical points which need to be polished, though. see details below.

# Details

* Betti numbers. I am sorry, I did not get your explanation on the

modified Edelsbrunner-Delfinado algorithm. on the other hand, if you

focus on closed - orientable surfaces, then, by the Thm of

classification for such surfaces, one has:

Euler char = num vertices - num edges + num triangles

= 2-2* genus = betti_0 (=1) - betti_1 + betti_2 ( = 1 ) = 2-betti_1.

Thus, computing the Betti numbers is trivial from the Euler char, which is trivial from the triangle mesh.

I also not that lines with large betti_1 means that the surface has

numerous tiny handles. These should be filtered out using topological

persistence -- see the Gudhi library.

* Curvatures: jet fitting versus MDSB. for Jet fitting, you need to

specify the degree of the jet and the number of neighbors. jet fitting

without these parameters does not make sense.

* Curvatures, sign. about << We have adopted the sign convention where

negative curvature values refer to convex regions.>>

you should be more precise: what the the relationship between

convexity and the local nature of the point (elliptic, hyperbolic,

parabolic) !???

* TetGen and constrained Delaunay. the main text should mention the

fact that GetGen uses a constrained Delaunay triangulation.

* Contenders and reproducibility of results, Fig. 7. one reads in the

answers that " We found that for some codes such as VolRoverN, Hu et

al. Remesh, and CGAL Mesh 3, many input meshes would produce

segmentation fault."

Since CGAL is supposed to be robust, we face two alternatives here:

either the program has not been compiled and/or user properly. (e.g.,

certain aggressive optimization options may jeopardize the numerics);

or there is a bug.

To clarify the situation, the supporting material should contain the

protocol used, and provide the files where the tests failed.

**Have all data underlying the figures and results presented in the manuscript been provided?**

Reviewer #1: Yes

Reviewer #2: Yes

Reviewer #4: Yes

PLOS authors have the option to publish the peer review history of their article (what does this mean?). If published, this will include your full peer review and any attached files.

Reviewer #1: No

Reviewer #2: No

Reviewer #4: No
---

## [Editor Report · Decision Letter 2]

1 Mar 2020

Dear Dr. Rangamani,

We are pleased to inform you that your manuscript '3D mesh processing using GAMer 2 to enable reaction-diffusion simulations in realistic cellular geometries' has been provisionally accepted for publication in PLOS Computational Biology.

Best regards,

Hugues Berry

Associate Editor

PLOS Computational Biology

Mona Singh

Methods Editor

PLOS Computational Biology

---

## [Editor Report · Acceptance letter]

18 Mar 2020

PCOMPBIOL-D-19-01230R2 

3D mesh processing using GAMer 2 to enable reaction-diffusion simulations in realistic cellular geometries

Dear Dr Rangamani,

I am pleased to inform you that your manuscript has been formally accepted for publication in PLOS Computational Biology. Your manuscript is now with our production department and you will be notified of the publication date in due course.

With kind regards,

Matt Lyles
